

# Asymptotic symmetries in the $TsT/T\bar{T}$ correspondence

**Zhengyuan Du**[1,2], **Wen-Xin Lai**[1,2,3], **Kangning Liu**[1,2] and **Wei Song**[1,2]

**1** Yau Mathematical Sciences Center, Tsinghua University, Beijing 100084, China
**2** Department of Mathematical Sciences, Tsinghua University, Beijing 100084, China
**3** Kavli Institute for Theoretical Sciences, University of Chinese Academy of Sciences, Beijing 100190, China

## Abstract

Starting from holography for IIB string theory on AdS$_3 \times \mathcal{N}$ with NS-NS flux, the TsT/$T\bar{T}$ correspondence is a conjecture that a TsT transformation on the string theory side is holographically dual to the single-trace version of the $T\bar{T}$ deformation on the field theory side. More precisely, the long string sector of string theory on the TsT-transformed background corresponds to the symmetric product theory whose seed theory is the $T\bar{T}$-deformed CFT$_2$. In this paper, we study the asymptotic symmetry of the string theory in the bulk. We find a state-dependent, non-local field redefinition under which the worldsheet equations of motion, stress tensor, as well as the symplectic form of string theory after the TsT transformation are mapped to those before the TsT transformation. The asymptotic symmetry in the auxiliary AdS basis is generated by two commuting Virasoro generators, while in the TsT transformed basis is non-linear and non-local. Our result from string theory analysis is compatible with that of the $T\bar{T}$ deformed CFT$_2$.

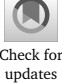

# 1 Introduction

The TsT/$T\bar{T}$ correspondence [1, 2] is a tractable toy model of holographic duality beyond the AdS/CFT correspondence constructed in string theory. The duality can be constructed by deforming an example of the AdS$_3$/CFT$_2$ correspondence from both sides. Before the deformation, the bulk theory is IIB string theory on AdS$_3 \times \mathcal{N}^7$ supported by NS-NS flux with electric charge $N$ and magnetic charge $k$. The background admits a weakly coupled string worldsheet description via the WZW model, the spectrum of which contains a short string sector with discrete representation and a long string sector with a continuum [3]. For superstring theory with $k = 1$ or bosonic string with $k = 3$, the short string sector disappears and the continuum is truncated so that the full spectrum is still discrete. In this case, the holographic dual theory is given by the symmetric product CFT denoted by $\mathcal{M}^N/S_N$ [4,5].[1] For generic values of $k$, the spectrum of the long string sector can still be matched with a symmetric product of Liouville CFT [7], whereas the full holographic theory requires a marginal deformation in order to incorporate the short string sector [8–10]. The TsT/$T\bar{T}$ correspondence [2, 11–13] deforms the aforementioned example of AdS$_3$/CFT$_2$ correspondence by a TsT transformation in the bulk string theory, and a single-trace $T\bar{T}$ deformation on the dual CFT$_2$ side.

On the boundary side, the single-trace $T\bar{T}$ deformation [1] of a symmetric product CFT $\mathcal{M}^N/S_N$ is also a symmetric product $\mathcal{M}^N_{T\bar{T}}/S_N$, where the seed theory $\mathcal{M}_{T\bar{T}}$ is the usual $T\bar{T}$ deformation [14–16] of the seed CFT $\mathcal{M}$. So far it is not clear how to define a single-trace $T\bar{T}$ deformation in the full spacetime CFT at a generic value of $k$, although the existence of such a deformation is expected. On the bulk side, the holographic dual is related to strings on some linear dilaton background, which can be described by a current-current deformation of the WZW model [17], and more generally by the TsT-transformed backgrounds [2]. TsT transformations are solution-generating techniques in supergravity, which can be used to generate new string backgrounds that are not asymptotically AdS or locally AdS. In higher dimensions, TsT transformations have been shown to be holographically dual to non-commutative, dipole, or $\beta$ deformations [18, 19]. The connection between TsT transformations and solvable irrelevant deformations of CFT$_2$ was first observed in the example of warped AdS$_3$ spacetime and single-trace $J\bar{T}$ deformation [11], generalized to the $O(d, d)$ deformations [12, 13], and systematically studied in [2, 20].

The TsT/$T\bar{T}$ correspondence provides a tractable model of flat holography in three spacetime dimensions with linear dilaton. The spectrum of the long string sector can be shown to match that of the single-trace $T\bar{T}$ deformed CFT, both in the untwisted sector [1, 2] and in the twisted sector [21]. A family of solutions containing both the black hole solutions and the smooth solution dual to the NS-NS ground state have been constructed, where the entropy and the gravitational charges of black holes can be reproduced by the single-trace $T\bar{T}$ deformed CFTs [2, 20], see also [22–25]. The partition function from string theory calculation [26] and from field theory calculation [21] are compatible with each other. See also [27] for interesting discussions of S-duality and UV completion of the theory by studying the partition sum. Due to the irrelevant nature of the $T\bar{T}$ deformation, the calculation of the correlation functions has been challenging, with perturbative results in [28–32], and a non-perturbative flow equation and Callan-Symanzik equation in [33, 34]. More recently, progress on non-perturbative calcu-

---

[1]See also [6] for an interpretation of the holographic dual theory as a grand canonical ensemble of free symmetric product CFTs. In this paper, we mainly focus on the string worldsheet theory and the different interpretations of the holographic dual do not affect subsequent discussions.

lations of the correlation functions in momentum space has been made both from the string theory side [35] and from the field theory side [36], the results of which are compatible in the high momentum limit. With a certain choice of normalization, two-point functions in the momentum space can be obtained from the CFT ones by a momentum-dependent shift of the conformal weights. This strongly suggests the possibility of finding underlying Virasoro symmetries, albeit non-local, in both the bulk and the boundary in the TsT/$T\bar{T}$ correspondence. This has been shown to be indeed the case in the single-trace $T\bar{T}$ deformed CFT$_2$ [37], a result which is based on previous work on double trace $T\bar{T}$ deformations [38]. In the bulk, we expect to find the asymptotic symmetry to have the same structure, which is the main focus of this paper.

In this paper, we further explore the TsT/$T\bar{T}$ correspondence by studying the asymptotic symmetries of the bulk string theory after the TsT transformation. The notion of asymptotic symmetry is crucial for a rigorous definition of conserved quantities such as energy in a theory of gravity. It also plays an important role in the bottom-up approach of holographic duality. The coincidence between the asymptotic symmetry on AdS$_3$ spacetime [39] and the conformal group in two dimensions indicates the potential existence of the AdS$_3$/CFT$_2$ correspondence. The discovery of BMS group [40–43] in asymptotically flat spacetime has also fostered the recent development of celestial holography, reviews of which can be found in e.g. [44–46]. Assuming the asymptotic Killing vectors found from the analysis of Einstein gravity, generators of the asymptotic symmetry for AdS$_3$ spacetime can be written as vertex operators on the worldsheet theory [47–49]. In [50], it is further observed that the boundary conditions imposed on the spacetime fields can be interpreted as falloff conditions on the worldsheet equations of motion and constraints. This provides a way of directly finding the asymptotic symmetries from the worldsheet theory. In this paper, we apply this method to the TsT/$T\bar{T}$ correspondence. A useful feature of TsT transformation is that a non-local field redefinition can map both the equations and the stress tensor after the transformation to those before [19]. This map, however, does not preserve the boundary conditions of the worldsheet fields. In section 4, we will further introduce a state-dependent nonlocal rescaling to restore the correct boundary conditions. Under the combined non-local field redefinition (53) with some specific integration constants (65), the equations of motion, stress tensor, boundary conditions, as well as the symplectic form of the string theory after the TsT transformation are mapped to those before the TsT deformation, the latter of which is referred to as the auxiliary $AdS$ string theory. There will then be two natural sets of variables: those in the TsT transformed theory and those in the auxiliary $AdS$ string theory. The asymptotic symmetry in the auxiliary $AdS$ basis is generated by two commuting Virasoro generators, while in the TsT transformed basis is non-linear and non-local. The result in this paper is consistent with the correlation functions [35, 36], symmetries of the $T\bar{T}$ deformation [38], as well as the perturbative analysis of asymptotic symmetry in supergravity [51].

The layout of this paper is as follows: in section 2 we review the basic setup of the TsT/$T\bar{T}$ correspondence, in section 3 we review asymptotic symmetries for string theory on AdS$_3$, in section 4 we discuss the nonlocal map which relates string theories before and after the TsT transformation, and in section 5 we discuss asymptotic symmetries for the TsT transformed string theory.

## 2 The TsT/$T\bar{T}$ correspondence

The long-string sector of string theory on the TsT transformed AdS$_3$ background shares many similar features with the single-trace $T\bar{T}$ deformation of the boundary CFT$_2$.[2] Here we will briefly review the key ingredients of the holographic dictionary, mostly following the conventions of [2,35].

The TsT transformations can be defined for any string theory background with two $U(1)$ isometries [18]. Let us denote the undeformed $U(1) \times U(1)$ directions as $(\tilde{x}^1, \tilde{x}^{\bar{2}})$. TsT means that we first perform T-duality along the $\tilde{x}^1$ circle, then shift $\tilde{x}^{\bar{2}}$ to $x^{\bar{2}}$ by mixing with $x^1$, namely $\tilde{x}^{\bar{2}} = x^{\bar{2}} - 2\lambda x^1/k$, and finally carry out T-duality along $x^1$ again. For nonzero $\lambda$ this leads to new supergravity backgrounds with new $U(1) \times U(1)$ coordinates $(x^1, x^2)$, due to the nontrivial shift sandwiched between the two T-dualities. Crucially, it has been observed that the TsT transformation can be realized on the worldsheet by a current-current deformation parametrized by $\lambda$:

$$\frac{\partial S_\lambda}{\partial \lambda} = -\frac{1}{\pi k} \int j \wedge \bar{j}, \tag{1}$$

where $j$ and $\bar{j}$ are worldsheet current 1-forms associated with the two $U(1)$ symmetries of translation in the target space, and $k$ is the number of NS5 branes generating the undeformed AdS$_3$ background. Note that $j$ and $\bar{j}$ on the right-hand side are $U(1)$ currents of the deformed theory at parameter $\lambda$, and thus (1) should be understood as a differential equation for the flow of worldsheet action. The deformation is expected to preserve these two $U(1)$ symmetries along the flow, and to be exactly marginal on the worldsheet. We will now focus on type IIB string theory on AdS$_3$ with pure NS-NS flux, which features two $U(1)$ null directions, here denoted as $(\tilde{u}, \tilde{v})$. These are also the coordinates of the dual CFT$_2$. Let us now restrict to the long string sector in this background, the spectrum of which coincides with a symmetric orbifold $\mathcal{M}^N/S_N$, where $\mathcal{M}$ is the seed CFT which contains a Liouville part [7]. For the $a$-th copy in the symmetric product, the boundary symmetry currents corresponding to the $(\tilde{u}, \tilde{v})$ shift symmetries are

$$\begin{aligned} J^a &= T^a_{xi} dx^i = T^a_{xx} dx + T^a_{x\bar{x}} d\bar{x}, \\ \bar{J}^a &= T^a_{\bar{x}i} dx^i = T^a_{\bar{x}x} dx + T^a_{\bar{x}\bar{x}} d\bar{x}. \end{aligned} \tag{2}$$

It would be natural to assume that the TsT transformed AdS$_3$, generated by the current-current deformation as in (1), would correspond to some deformation with a similar structure on the boundary CFT$_2$. Indeed, the worldsheet deformation (1) corresponds to a deformation summing over each seed theory $\mathcal{M}$ of the symmetric orbifold:

$$\frac{\partial S_\mu}{\partial \mu} = -\frac{1}{\pi} \sum_{a=1}^N \int J^a \wedge \bar{J}^a. \tag{3}$$

The integrand $J^a \wedge \bar{J}^a$ is proportional to the stress tensor determinant $\det T^a_{ij}$, so this is precisely the $T\bar{T}$ deformation [14–16] on the $a$-th seed theory. The full deformation is obtained by summing over the index $a = 1, \cdots, N$, which leads to the single-trace $T\bar{T}$ deformation on the dual field theory side.

A crucial evidence for the TsT/$T\bar{T}$ correspondence is the agreement of the deformed spectrum on a cylinder of radius $R$:

$$E(\mu) = -\frac{wR}{2\mu}\left[1 - \sqrt{1 + \frac{4\mu}{wR}E(0) + \frac{4\mu^2}{w^2 R^4}J(0)^2}\right], \qquad J(\mu) = J(0), \tag{4}$$

---

[2]As the string theory in the bulk also contains the short string sector, the dual field theory is not a symmetric product theory even before the deformation. Nevertheless we expect that the full theory of the deformed CFT, although not been precisely defined so far, still share some similar features of the single-trace $T\bar{T}$ deformation.

where $w$ labels the $w$-twisted sector of the symmetric orbifold at the boundary, which corresponds to the winding number of a long string in the bulk. The deformed spectrum in the twisted sector can be independently obtained from the field theory side with the single-trace $T\bar{T}$ deformation [21], and from the string theory side with worldsheet analysis [2,17], if we identify the parameters:

$$\lambda = \ell_s^{-2}\mu, \qquad \ell = R. \tag{5}$$

The fact that the deformed spectrum is solvable suggests strongly that the deformed theory is constrained by additional symmetries. Field theoretic and supergravity analysis of symmetries in $T\bar{T}$-deformed CFTs have been previously discussed in e.g. [37,38,51–53]. In this paper we will attack the problem from the perspective of worldsheet string theory (1).

## 3 Asymptotic symmetry from the worldsheet theory

In this section, we explain the strategy of studying asymptotic symmetry from the string worldsheet proposed in [50], and review the relevant results on string theory on $\text{AdS}_3 \times \mathcal{N}$ with NS-NS flux.

### 3.1 Asymptotic symmetry from the worldsheet theory

In a usual quantum field theory without gravity, translational symmetry and Lorentzian invariance are continuous global symmetries, which according to Noether's theorem are generated by conserved charges. In a theory containing gravity, gravitational charges can be similarly defined using the Noether procedure after specifying the boundary conditions [54], under which diffeomorphisms are classified into three types: large, small, and forbidden. Forbidden diffeomorphisms violate the boundary conditions and hence are not allowed. Small diffeomorphisms fall off fast near the boundary and are trivial gauge redundancies. The most interesting ones are large diffeomorphisms which preserve the boundary conditions but have a non-trivial effect at the boundary. Due to the boundary conditions, large diffeomorphisms are no longer gauge redundancies, but instead symmetry transformations that map states to states in the Hilbert space. The asymptotic symmetry group is formed by these large diffeomorphisms.

For Einstein gravity with negative cosmological constant in three dimensions, Brown and Henneaux [39] found consistent boundary conditions under which the asymptotic group is generated by left and right-moving Virasoro generators. To describe IIB string theory on $\text{AdS}_3 \times \mathcal{N}$ with NS-NS flux, the three-dimensional gravity has to also include a dilaton and a Kalb-Ramond 2-form field. Under the boundary conditions [50], it is found that Virasoro generators are accompanied by a large gauge transformation of the 2-form field. Nevertheless, the resulting conserved charges and the asymptotic group remain the same as in pure Einstein gravity.

Now let us consider asymptotic symmetries on the string worldsheet. In the WZW model which describes the three-dimensional part of IIB string theory on $\text{AdS}_3 \times \mathcal{N}$ with NS-NS flux, vertex operators [47,48,55] on the worldsheet have been written down as the Virasoro generators in the target spacetime. It is shown in [50] that the asymptotic Killing vectors can be directly worked out by requiring that the worldsheet equations of motion and constraints are satisfied near the asymptotic boundary in the target spacetime. Symmetry generators on the worldsheet are then interpreted as Noether charges. Asymptotic symmetries on the string worldsheet for flat spacetime have been discussed in [50,56–58]. In the following, we explain the main steps of finding the asymptotic symmetries on the worldsheet in [50].

**The asymptotic Killing vectors**

Consider the bosonic part of worldsheet action of string theory in the conformal gauge with target spacetime metric $G_{\mu\nu}$ and Kalb-Ramond field $B_{\mu\nu}$,

$$S = \frac{1}{4\pi\alpha'} \int d^2\sigma \, M_{\mu\nu}\partial X^\mu \bar\partial X^\nu, \qquad M_{\mu\nu} = G_{\mu\nu} + B_{\mu\nu}. \tag{6}$$

Given a specific background $M_{\mu\nu}$, a spacetime diffeomorphism,

$$\delta_\xi X^\mu = \xi^\mu, \tag{7}$$

is an asymptotic symmetry if the worldsheet equations of motion and stress tensor are preserved near the boundary[3]

$$\delta_\xi\left(\bar\partial(M_{\mu\lambda}\partial X^\mu) + \partial(M_{\lambda\nu}\bar\partial X^\nu) - \partial_\lambda M_{\mu\nu}\partial X^\mu \bar\partial X^\nu\right) \to 0,$$
$$\delta_\xi T_{ws} \to 0, \qquad \delta_\xi \bar{T}_{ws} \to 0. \tag{8}$$

These conditions will in principle enable us to solve for the asymptotic Killing vectors $\xi$. The generators of the asymptotic symmetry can be written down either in the Lagrangian formalism or in the Hamiltonian formalism.

**Charges in the Lagrangian formalism**

To derive the Noether charge in the Lagrangian formalism, we note that the variation of the action under a diffeomorphism $\epsilon(z,\bar{z})\xi^\mu$ and background gauge transformation $\delta_{\epsilon\Lambda}B_{\mu\nu} = \partial_\mu(\epsilon\Lambda_\nu) - \partial_\nu(\epsilon\Lambda_\mu)$ is given by

$$\delta_{\epsilon\xi,\epsilon\Lambda}S = \frac{1}{2\pi}\int d^2z\left(\epsilon V + \partial\epsilon\, j_{\bar{z}} + \bar\partial\epsilon\, j_z\right),$$
$$j_z = \frac{1}{\alpha'}(\xi^\nu M_{\mu\nu} - \Lambda_\mu)\partial X^\mu, \qquad j_{\bar{z}} = \frac{1}{\alpha'}(\xi^\mu M_{\mu\nu} + \Lambda_\nu)\bar\partial X^\nu, \tag{9}$$
$$V = \frac{1}{\alpha'}\left(\mathscr{L}_\xi M_{\mu\nu} + \partial_\mu\Lambda_\nu - \partial_\nu\Lambda_\mu\right)\partial X^\mu \bar\partial X^\nu,$$

which after using the equations of motion satisfies the divergence law

$$\bar\partial j_z + \partial j_{\bar{z}} = V. \tag{10}$$

If we can find $\Lambda_\mu$ so that the vertex $V$ vanishes on-shell at the boundary, the Noether charge is then given by

$$J = \frac{1}{2\pi}\left(\oint dz\, j_z - \oint d\bar{z}\, j_{\bar{z}}\right). \tag{11}$$

In [50], it is shown that spacetime Virasoro generators in the $SL(2,\mathbb{R})$ WZW model and $\mathrm{BMS}_3$ generators in string theory on three-dimensional flat space can both be derived using this procedure. In particular, the large gauge transformation is necessary for the vertex to vanish asymptotically.

---

[3]The falloff should be further specified in explicit examples.

## Charges in the Hamiltonian formalism

Now let us consider charges in the Hamiltonian formalism in a phase space parameterized by $q^I \in \{x^\mu, p_\mu, \mu = 1, \cdots d\}$, with the canonical symplectic structure

$$\omega = \frac{1}{2}\omega_{IJ}\delta q^I \wedge \delta q^J, \tag{12}$$

where $\omega_{IJ}$ are independent of $q^I$, $x^\mu$ are the coordinates of the target spacetime and $p_\mu$ are the momenta. Suppose a translation in the phase space along $\delta_\xi q^I \equiv \xi^I$ is generated by the charge $H_\xi$, then for an arbitrary functional $P$ of $q^I$, we have

$$\delta_\xi P \equiv \xi^I \frac{\delta P}{\delta q^I} = \{P, H_\xi\} = \omega^{IJ}\frac{\delta P}{\delta q^I}\frac{\delta H_\xi}{\delta q^J}, \tag{13}$$

where $\omega^{IJ}$ is the inverse of $\omega_{IJ}$. The above equation implies the relation

$$\xi^I = \omega^{IJ}\frac{\delta H_\xi}{\delta q^J}, \tag{14}$$

which further allows us to derive the infinitesimal charge defined near a point in the phase space as

$$\delta H_\xi \equiv \frac{\delta H_\xi}{\delta q^I}\delta q^I = -\xi^K\omega_{KJ}\delta q^J. \tag{15}$$

For a consistent choice of the tangent vector $\xi^I$ in the phase space satisfying (14), the infinitesimal charge $\delta H_\xi$ is a closed 1-form in the phase space and thus should be integrable. Therefore charge integrability can be used as a consistent condition for $\xi^I$.

For the purpose of discussing asymptotic symmetries on the worldsheet theory, we can determine the phase space vector $\xi^I$ from its components in the spacetime coordinates $\xi^\mu = \delta_\xi x^\mu$, $\mu = 1, \cdots d$, following the procedure proposed in [50]. For a given spacetime diffeomorphism $\xi^\mu = \{x^\mu, H_\xi\}$, we can determine the variation of the momentum by requiring the following conditions

$$\delta_\xi H = \{H, H_\xi\} \rightarrow 0,$$
$$\{\xi^I, H\} - \{\{q^I, H\}, H_\xi\} = \{q^I, \{H_\xi, H\}\} \rightarrow 0, \qquad q^I \in \{x^\mu, p_\nu\}, \tag{16}$$

where the arrow denotes the limit as it approaches the asymptotic boundary. Explicit falloff conditions will be further specified in different examples. The first condition in (16) indicates that the Hamiltonian is preserved by the transformation generated by $H_\xi$ in the asymptotic region, or equivalently the charge $H_\xi$ is asymptotically preserved. The second equation in (16) is a combination of the Jacobi identity and the charge conservation condition, and the physical meaning is that the transformation $H_\xi$ is compatible with the Hamiltonian evolution and thus preserves the equations of motion asymptotically.

Solving the equations (16) for the vector $\xi^I$, and plugging the solutions into (15), we get the infinitesimal charge that generates transformation $\xi^I$ in the phase space, which if integrable, can be further integrated to obtain the finite charge $H_\xi$. In [50], this procedure has been used to derive the charges that generate asymptotic symmetries of the $SL(2, \mathbb{R})$ WZW model and string theory on three-dimensional flat spacetime. In this paper, we will further carry out the analysis of the string worldsheet theory obtained from the TsT transformation of the WZW model.

## 3.2 IIB string theory on AdS$_3$ × $\mathcal{N}$

The three dimensional part of IIB string theory on asymptotically AdS$_3$ × $\mathcal{N}$ background with NS-NS background can be described by the $SL(2,\mathbb{R})$ WZW model, a theory that has been studied extensively in the literature. The spectrum [3,59,60] contains both the long string sector and the short string sector. For superstring with NS5-brane charge $k = 1$, or bosonic string with $k = 3$, it has been demonstrated that the holographic dual is given by a symmetric product CFT [5]. For generic $k$, while the long string sector can still be holographically described by a symmetric product CFT [7], the symmetric product structure is necessarily broken [8–10] in order to include the short string sector.

We are interested in the asymptotic symmetry. For that purpose, it is convenient to consider cylindrical boundaries, a setup where Brown-Henneaux boundary conditions [39] were imposed in pure Einstein gravity. The phase space is usually described by the Bañados metrics in the Fefferman-Graham gauge and contains the global AdS$_3$ and BTZ black holes. In particular, the string background with a non-rotating BTZ background with zero mass can be written in the string frame by

$$
\begin{aligned}
d\tilde{s}^2 &= \ell^2 \left\{ d\tilde{\phi}^2 + \exp(2\tilde{\phi})\, d\tilde{u}\, d\tilde{v} \right\}, \qquad (\tilde{u}, \tilde{v}) \sim (\tilde{u} + 2\pi, \tilde{v} + 2\pi), \\
\tilde{B}_{\mu\nu} &= -\frac{\ell^2}{2} \exp(2\tilde{\phi})\, d\tilde{u} \wedge d\tilde{v}, \\
e^{2\tilde{\Phi}} &= \frac{k}{N} e^{-2\phi_0}, \quad k = \ell^2/\ell_s^2,
\end{aligned}
\tag{17}
$$

where we have omitted the internal spacetime, and used the lightcone coordinates $\tilde{u} := \tilde{\varphi} + \tilde{t}$ and $\tilde{v} := \tilde{\varphi} - \tilde{t}$. The magnetic charge $k = \ell^2/\ell_s^2$ specifies how large the curvature scale is compared to the string scale. A small value of $k$ indicates strong stringy effects. $N$ is the electric charge, which is assumed to be large. Using the plane coordinate on the worldsheet with $z := \exp(i(\sigma - i\tau))$ and $\bar{z} := \exp(-i(\sigma + i\tau))$, the string worldsheet theory on (17) can be written in the conformal gauge as

$$
\tilde{S} = \frac{1}{4\pi\alpha'} \int d^2z\, \tilde{M}_{\mu\nu} \partial\tilde{x}^\mu \bar{\partial}\tilde{x}^\nu = \frac{k}{2\pi} \int d^2z \left\{ \partial\tilde{\phi}\bar{\partial}\tilde{\phi} + \exp(2\tilde{\phi})\bar{\partial}\tilde{u}\partial\tilde{v} \right\},
\tag{18}
$$

where $d^2z = dz\, d\bar{z}$. The stress tensor is

$$
T_{ws} = -k\, \partial\phi\partial\phi - k \exp(2\phi)\, \partial u\partial v.
\tag{19}
$$

At the quantum level, the level of the WZW model acquires a shift and the action reads [47, 61,62]

$$
\tilde{S} = \frac{1}{2\pi} \int dz^2 \left\{ (k-2)\partial\tilde{\phi}\bar{\partial}\tilde{\phi} + k \exp(2\tilde{\phi})\bar{\partial}\tilde{u}\partial\tilde{v} - \frac{1}{4}\tilde{\phi}R_{ws} \right\},
\tag{20}
$$

where $R_{ws}$ is the worldsheet curvature which vanishes on a flat worldsheet metric. Throughout this paper, we only focus on flat worldsheets where the last term in (20) does not play a role except for deriving the stress tensor, the latter of which is given by

$$
\tilde{T}_{ws} = -(k-2)\, \partial\tilde{\phi}\partial\tilde{\phi} - k \exp(2\tilde{\phi})\partial\tilde{u}\partial\tilde{v} - \partial^2\tilde{\phi}.
\tag{21}
$$

The background (17) is invariant under translations along $u$ and $v$, which are generated by the Noether currents on the worldsheet,

$$
\tilde{j}_0 = k \exp(2\tilde{\phi})\, \partial\tilde{v}, \qquad \tilde{\bar{j}}_0 = k \exp(2\tilde{\phi})\, \bar{\partial}\tilde{u},
\tag{22}
$$

with Noether charges

$$\tilde{J}_0 := -\frac{1}{2\pi} \oint dz \tilde{j}_0(z), \qquad \tilde{\bar{J}}_0 := -\frac{1}{2\pi} \oint d\bar{z} \tilde{\bar{j}}_0(\bar{z}).$$

(23)

The worldsheet equations of motion can be written as

$$(k-2)\partial\bar{\partial}\tilde{\phi} - k\exp(2\tilde{\phi})\bar{\partial}\tilde{u}\partial\tilde{v} = 0,$$
$$\bar{\partial}\tilde{j}_0 = \partial\tilde{\bar{j}}_0 = 0,$$

(24)

where the second line is just the conservation law for the two $U(1)$ currents (22). The OPEs in the large $\phi$ limit is given by,

$$\tilde{\phi}(z,\bar{z})\tilde{\phi}(w,\bar{w}) \sim -\frac{1}{2(k-2)}\log|z-w|^2,$$
$$\tilde{j}_0(z)\tilde{u}(w) \sim -\frac{1}{z-w}, \qquad \tilde{\bar{j}}_0(\bar{z})\tilde{v}(\bar{w}) \sim -\frac{1}{\bar{z}-\bar{w}}.$$

(25)

**Asymptotic symmetries for strings on AdS$_3$**

As explained in [50] and summarized in section 3.1, asymptotic Killing vectors can be determined by requiring the variation of the worldsheet equation of motion to vanish up to specific orders at the boundary. For massless BTZ, we impose the following boundary conditions on the equations of motion,

$$(k-2)\partial\bar{\partial}\tilde{\xi}^\phi - 2k\tilde{\xi}^\phi\exp(2\tilde{\phi})\bar{\partial}\tilde{u}\partial\tilde{v} - k\exp(2\tilde{\phi})\bar{\partial}\tilde{\xi}^u\partial\tilde{v} - k\exp(2\tilde{\phi})\bar{\partial}\tilde{u}\partial\tilde{\xi}^v = \mathcal{O}(\exp(-4\tilde{\phi})),$$
$$\bar{\partial}\left(\exp(2\tilde{\phi})\partial\xi^v + 2\xi^\phi\exp(2\tilde{\phi})\partial\tilde{v}\right) = \mathcal{O}(\exp(-2\tilde{\phi})),$$
$$\partial\left(\exp(2\tilde{\phi})\bar{\partial}\xi^u + 2\xi^\phi\exp(2\tilde{\phi})\bar{\partial}\tilde{u}\right) = \mathcal{O}(\exp(-2\tilde{\phi})).$$

(26)

In addition, we note that finiteness of the currents (22) implies that $u$ is asymptotically chiral and $v$ is anti-chiral. To preserve this property, we need to impose the chirality condition on the asymptotic Killing vector,

$$\bar{\partial}\tilde{\xi}^u = \mathcal{O}(\exp(-2\tilde{\phi})), \qquad \partial\tilde{\xi}^v = \mathcal{O}(\exp(-2\tilde{\phi})).$$

(27)

Solving the asymptotic on-shell condition and chirality condition, we obtain the Brown-Henneaux asymptotic Killing vectors [39]

$$\tilde{\xi} = \tilde{\xi}^u\partial_{\tilde{u}} + \tilde{\xi}^v\partial_{\tilde{v}} + \tilde{\xi}^\phi\partial_{\tilde{\phi}},$$

(28)

where

$$\tilde{\xi}^u = f(\tilde{u}) - \frac{k-2}{2k}\exp(-2\tilde{\phi})\bar{f}''(\tilde{v}) + \mathcal{O}(\exp(-4\tilde{\phi})),$$
$$\tilde{\xi}^v = \bar{f}(\tilde{v}) - \frac{k-2}{2k}\exp(-2\tilde{\phi})f''(\tilde{u}) + \mathcal{O}(\exp(-4\tilde{\phi})),$$
$$\tilde{\xi}^\phi = -\frac{1}{2}f'(\tilde{u}) - \frac{1}{2}\bar{f}'(\tilde{v}) + \mathcal{O}(\exp(-2\tilde{\phi})).$$

(29)

The above procedure can also be carried out for all the Bañados metrics. In the Ferfferman-Graham gauge, we will obtain the same asymptotic on-shell condition (26) and chirality conditions (27). As a consequence, we will find the same asymptotic Killing vectors (29).

The Noether charges that generate the above asymptotic symmetry transformation can be written down using (9), where the gauge parameter can be determined by requiring the vertex to vanish. For AdS$_3$, the Noether current for the symmetry parameterized by $f(\tilde{u})$ is given by

$$
\begin{aligned}
\tilde{j}_z &= k f(\tilde{u}) \exp(2\tilde{\phi}) \partial \tilde{v} - (k-2) f'(\tilde{u}) \partial \tilde{\phi}, & \tilde{j}_{\bar{z}} &= -\frac{k-2}{2} f''(\tilde{u}) \bar{\partial} \tilde{u}, \\
\tilde{\bar{j}}_{\bar{z}} &= k \bar{f}(\tilde{v}) \exp(2\tilde{\phi}) \bar{\partial} \tilde{u} - (k-2) \bar{f}'(\tilde{v}) \bar{\partial} \tilde{\phi}, & \tilde{\bar{j}}_z &= -\frac{k-2}{2} \bar{f}''(\tilde{v}) \partial \tilde{v},
\end{aligned}
\tag{30}
$$

and the Noether charges are given by

$$
\tilde{J}_f = \frac{1}{2\pi} \left( \oint dz\, \tilde{j}_z - \oint d\bar{z}\, \tilde{j}_{\bar{z}} \right), \qquad \tilde{\bar{J}}_{\bar{f}} = \frac{1}{2\pi} \left( -\oint d\bar{z}\, \tilde{\bar{j}}_{\bar{z}} + \oint dz\, \tilde{\bar{j}}_z \right).
\tag{31}
$$

For completeness, we have kept the anti-chiral component $\tilde{j}_{\bar{z}}$, which is necessary to generate the $e^{-2\phi} f''(\tilde{u})$ term in (28). As this term is subleading, the current generating the transformation parameterized by $f(\tilde{u})$ is chiral near the asymptotic boundary.

The asymptotic Killing vectors (28) have to preserve the periodic identification $(\tilde{u}, \tilde{v}) \sim (\tilde{u} + 2\pi, \tilde{v} + 2\pi)$, which restricts $f(\tilde{u})$ to be a periodic function of $\tilde{u}$. One can expand the periodic functions in Fourier modes

$$
\tilde{f}_n = -\exp(in\tilde{u}), \qquad \tilde{\bar{f}}_n = \exp(-in\tilde{v}).
\tag{32}
$$

The charges $\tilde{J}_n \equiv \tilde{J}_{\tilde{f}_n}$ form left and right-moving Virasoro algebras

$$
\begin{aligned}
\left[ \tilde{J}_n, \tilde{J}_m \right] &= (n-m) \tilde{J}_{n+m} + \frac{c}{12} n^3 \delta_{n,-m}, \\
\left[ \tilde{\bar{J}}_n, \tilde{\bar{J}}_m \right] &= (n-m) \tilde{\bar{J}}_{n+m} + \frac{\bar{c}}{12} n^3 \delta_{n,-m}, \\
\left[ \tilde{J}_n, \tilde{\bar{J}}_m \right] &= 0,
\end{aligned}
\tag{33}
$$

where the central charges depend on the worldsheet topology and are given by

$$
c = \bar{c} = 6 k \mathcal{I}, \qquad \mathcal{I} = \frac{1}{2\pi} \oint dz\, \partial \tilde{u}.
\tag{34}
$$

Using the OPE (25), we obtain the following OPE between the spacetime Virasoro current and the worldsheet stress tensor

$$
\tilde{T}_{ws}(z) \tilde{j}_z(w) = \frac{\tilde{j}_z(w)}{(z-w)^2} + \frac{\partial \tilde{j}_z(w)}{z-w} + \cdots
\tag{35}
$$

This means that the left-moving spacetime Virasoro currents are worldsheet primary operators with conformal weight $(1,0)$, and similarly the right-moving Virasoro currents have weights $(0,1)$. Performing the contour integral, we find that the spacetime Virasoro transformations leave the worldsheet stress tensor invariant asymptotically,

$$
[\tilde{J}_m, T_{ws}] = [\tilde{\bar{J}}_m, T_{ws}] = 0,
\tag{36}
$$

and thus are indeed asymptotic symmetries in the sense that they map physical states among themselves.

# 4 TsT transformation and the nonlocal map

In this section, we describe TsT transformations and discuss a non-local field redefinition that maps string theories before and after the TsT transformation. We show that such a field redefinition can be understood as a canonical transformation of the worldsheet symplectic structure.

## 4.1 TsT transformation on the string worldsheet

Starting from type IIB string theory on the AdS$_3$ background (17), we perform a TsT deformation by T-duality along $\tilde{u}$, shifting $\tilde{v} \to \tilde{v} - \frac{2\lambda}{k}\tilde{u}$ and T-duality along $\tilde{u}$ again. The TsT-transformed combination $M_{\mu\nu} = G_{\mu\nu} + B_{\mu\nu}$ can be obtained from the undeformed one by a relation [2,18],

$$M = \tilde{M}\left(I + \frac{2\lambda}{\ell^2}\Gamma\tilde{M}\right)^{-1}, \qquad \Phi = \tilde{\Phi} + \frac{1}{4}\log\frac{\det G_{\mu\nu}}{\det \tilde{G}_{\mu\nu}}, \tag{37}$$

where $\Gamma_{\mu\nu} = \delta_\mu^u \delta_\nu^v - \delta_\mu^v \delta_\nu^u$ is a totally antisymmetric tensor along the $u$ and $v$ directions. This follows directly from the Buscher rules [63] of T-dualities. This leads to the new background:

$$ds^2 = \ell^2\left\{d\phi^2 + \frac{\exp(2\phi)}{1 + 2\lambda\exp(2\phi)}du\,dv\right\},$$
$$B = -\frac{\ell^2}{2}\frac{\exp(2\phi)}{1 + 2\lambda\exp(\phi)}du \wedge dv, \tag{38}$$
$$e^{2\Phi} = \frac{k}{N}\frac{1}{1 + 2\lambda\exp(2\phi)}e^{-2\phi_0}.$$

The string theory defined on this background is given by

$$S = \frac{k}{2\pi}\int d^2z\left\{\partial\phi\bar{\partial}\phi + \frac{\exp(2\phi)}{1 + 2\lambda\exp(2\phi)}\bar{\partial}u\partial v\right\}. \tag{39}$$

The quantum action can be obtained by a TsT transformation from (20) and is given by

$$S = \frac{1}{2\pi}\int d^2z\left\{(k-2)\partial\phi\bar{\partial}\phi + \frac{k\exp(2\phi)}{1 + 2\lambda\exp(2\phi)}\bar{\partial}u\partial v - \frac{1}{4}\phi R_{ws}\right\}. \tag{40}$$

In the classical limit with $k \to \infty$, the action (40) reduces to the classical one (39). We are interested in the massless BTZ background whose conformal boundary is a cylinder with the following identification,

$$(u, v) \sim (u + 2\pi, v + 2\pi). \tag{41}$$

The equations of motion from the action (40) are

$$(k-2)\partial\bar{\partial}\phi = \frac{k\exp(2\phi)}{(1 + 2\lambda\exp(2\phi))^2}\bar{\partial}u\partial v,$$
$$\partial j_0 = 0, \qquad \partial\bar{j}_0 = 0, \tag{42}$$

where

$$j_0 = k\frac{\exp(2\phi)}{1 + 2\lambda\exp(2\phi)}\partial v, \qquad \bar{j}_0 = k\frac{\exp(2\phi)}{1 + 2\lambda\exp(2\phi)}\bar{\partial}u, \tag{43}$$

are the worldsheet Noether currents generating translations along the target space coordinates $u$ and $v$. It is not difficult to see that the action (40) is an explicit solution of the worldsheet differential equation (1) where the currents are given by (43). The zero mode charges of these currents are left and right moving energies in spacetime,

$$J_0 := -\frac{1}{2\pi}\oint_t d\sigma j_0(\sigma) = -\frac{1}{2\pi}\oint dz j_0(z), \qquad \bar{J}_0 := -\frac{1}{2\pi}\oint_t d\sigma \bar{j}_0(\sigma) = -\frac{1}{2\pi}\oint d\bar{z}\bar{j}_0(\bar{z}). \tag{44}$$

Solutions to the equations of motion have to satisfy the boundary condition

$$u(\sigma + 2\pi) = u(\sigma) + 2\pi w, \qquad v(\sigma + 2\pi) = v(\sigma) + 2\pi w, \qquad w \in \mathbb{Z}, \tag{45}$$

where $w$ is the winding around the boundary circle (41). Physical states also need to satisfy the Virasoro constraints, where the worldsheet stress tensor is given by

$$T = -\left\{(k-2)\partial\phi\partial\phi + \frac{k\exp(2\phi)}{1+2\lambda\exp(2\phi)}\partial u\partial v + \partial^2\phi\right\},$$
$$\bar{T} = -\left\{(k-2)\bar{\partial}\phi\bar{\partial}\phi + \frac{k\exp(2\phi)}{1+2\lambda\exp(2\phi)}\bar{\partial}u\bar{\partial}v + \bar{\partial}^2\phi\right\}. \tag{46}$$

## 4.2 TsT as a field redefinition

As was explained in [2, 19], the worldsheet equations of motion and the stress tensor before and after the TsT transformation are related by the following field redefinition

$$\hat{\phi} = \phi,$$
$$\partial\hat{u} = \partial u, \qquad\qquad \bar{\partial}\hat{u} = \bar{\partial}u - \frac{2\lambda}{k}\bar{j}_0,$$
$$\partial\hat{v} = \partial v - \frac{2\lambda}{k}j_0, \qquad \bar{\partial}\hat{v} = \bar{\partial}v. \tag{47}$$

Let us define fields

$$\eta(z) \equiv \int^z dz'\, j_0(z') + \eta_0, \qquad \bar{\eta}(\bar{z}) \equiv \int^{\bar{z}} d\bar{z}'\, \bar{j}_0(\bar{z}') + \bar{\eta}_0, \tag{48}$$

where $\eta_0, \bar{\eta}_0$ are integration constants that may potentially depend on the state and will be discussed in detail momentarily. Then the field redefinition (47) can be written as

$$\hat{u} = u - \frac{2\lambda}{k}\bar{\eta}, \qquad \hat{v} = v - \frac{2\lambda}{k}\eta. \tag{49}$$

Under the above field redefinition, the $U(1)$ currents (43) after the TsT transformation become those on AdS$_3$ (22) with the tilded variables replaced by the hatted variables,

$$j_0(x^\mu) = \hat{j}_0(\hat{x}^\mu) = k\exp(2\hat{\phi})\partial\hat{v}, \qquad \bar{j}_0(x^\mu) = \hat{\bar{j}}_0(\hat{x}^\mu) = k\exp(2\hat{\phi})\bar{\partial}\hat{u}, \tag{50}$$

so that the equations of motion (42) after the TsT transformation are equivalent to those on the original $AdS_3 \times \mathcal{N}$ background,

$$(k-2)\partial\bar{\partial}\hat{\phi} = k\exp(2\hat{\phi})\bar{\partial}\hat{u}\partial\hat{v}, \qquad \bar{\partial}\hat{j}_0 = \partial\hat{\bar{j}}_0 = 0. \tag{51}$$

However, the boundary condition (45) implies that the hatted variables now satisfy the twisted boundary conditions,

$$\hat{u}(\sigma + 2\pi) = \hat{u}(\sigma) + 2\pi w R_u, \qquad R_u = 1 + \frac{2\lambda}{wk}\bar{J}_0,$$
$$\hat{v}(\sigma + 2\pi) = \hat{v}(\sigma) + 2\pi w R_v, \qquad R_v = 1 + \frac{2\lambda}{wk}J_0, \tag{52}$$

where $J_0$ and $\bar{J}_0$ are the charges (44) which generate translations along $u$ and $v$, respectively. The twisted boundary condition in $\hat{u}$ can be realized by a spectral flow transformation, using which the spectrum before and after the TsT transformation can be related [2, 11].[4] Note that the additional constants in the field redefinition (49) do not affect the boundary conditions (52). To discuss the symmetries, it is more convenient to introduce the following new

---

[4]The field redefinition (49) and the twisted boundary condition (52) are reminiscent of the state-dependent coordinate transformations in double-trace $T\bar{T}$ deformed CFTs [53, 64, 65].

variables, collectively denoted by $\hat{X}$, to absorb the twisted boundary conditions by a field-dependent rescaling transformation in the target spacetime,

$$\hat{\Phi} = \phi + \frac{1}{2}\log(R_u R_v),$$

$$\hat{U} = \frac{\hat{u}}{R_u} = \left(u - \frac{2\lambda}{k}\bar{\eta}\right)\frac{1}{R_u},$$

$$\hat{V} = \frac{\hat{v}}{R_v} = \left(v - \frac{2\lambda}{k}\eta\right)\frac{1}{R_v},$$

(53)

such that the $\hat{X}$ variables satisfy periodic boundary conditions,

$$\hat{U}(\sigma + 2\pi) = \hat{U}(\sigma) + 2\pi w, \qquad \hat{V}(\sigma + 2\pi) = \hat{V}(\sigma) + 2\pi w.$$

(54)

Note that the new spacetime coordinates $\hat{X}$ are only defined in a fixed winding sector. We restrict all subsequent discussions within this sector in the current paper. It is straightforward to see that the equations of motion (42) for the TsT coordinates $x^\mu \in \{u, v, \phi\}$ can be written in terms of the new variables $\hat{X}^\mu \in \{\hat{U}, \hat{V}, \hat{\Phi}\}$, the latter of which takes a similar form as the equations of motion of the tilded variables, i.e.

$$k e^{2\hat{\Phi}}\bar{\partial}\hat{U}\partial\hat{V} - (k-2)\partial\bar{\partial}\hat{\Phi} = 0, \qquad \bar{\partial}\mathcal{J}_0 = \partial\bar{\mathcal{J}}_0 = 0,$$

(55)

where the chiral current $\mathcal{J}_0$ and anti-chiral current $\bar{\mathcal{J}}_0$ are analogous to (22),

$$\mathcal{J}_0 \equiv k\exp(2\hat{\Phi})\partial\hat{V} = j_0 R_u,$$

$$\bar{\mathcal{J}}_0 \equiv k\exp(2\hat{\Phi})\bar{\partial}\hat{U} = \bar{j}_0 R_v.$$

(56)

The conservation law in (55) then allows us to define the conserved charges

$$\mathscr{J}_0 \equiv -\frac{1}{2\pi}\oint dz\,\mathcal{J}_0 = J_0 R_u,$$

$$\bar{\mathscr{J}}_0 \equiv -\frac{1}{2\pi}\oint d\bar{z}\,\bar{\mathcal{J}}_0 = \bar{J}_0 R_v,$$

(57)

where we have also worked out the relation between these charges and the two global $U(1)$ charges (44). Compared to the discussion in the WZW model, it is natural to guess that the charge $\mathscr{J}_0$ generates a translation of the non-local coordinate $\hat{U}$. As will be shown later, this is indeed true if we carefully choose the zero modes that appear in the field redefinition (53).

We have seen that the variables $\hat{X}$ satisfy the same equations of motion and boundary conditions as variables $\tilde{x}$ which are coordinates of AdS$_3$. Moreover, the stress tensor (46) can also be written in terms of the $\hat{X}$ variables, which does not explicitly depend on $\lambda$ and takes a similar form as the WZW model,

$$T = -\left\{(k-2)\partial\hat{\Phi}\partial\hat{\Phi} + k\exp(2\hat{\Phi})\partial\hat{U}\partial\hat{V} + \partial^2\hat{\Phi}\right\},$$

$$\bar{T} = -\left\{(k-2)\bar{\partial}\hat{\Phi}\bar{\partial}\hat{\Phi} + k\exp(2\hat{\Phi})\bar{\partial}\hat{U}\bar{\partial}\hat{V} + \bar{\partial}^2\hat{\Phi}\right\}.$$

(58)

This also implies that the worldsheet Hamiltonian is similar to that of string theory on AdS$_3$. To reproduce the equations of motion (55) and the stress tensor (58), the action for $\hat{X}^\mu$ is given by (20) with the tilded variables $\tilde{x}^\mu$ replaced by the upper-case hatted variables $\hat{X}^\mu$,

$$\hat{S} = \frac{1}{2\pi}\int d^2z\left\{(k-2)\,\partial\hat{\Phi}\bar{\partial}\hat{\Phi} + k\exp(2\hat{\Phi})\bar{\partial}\hat{U}\partial\hat{V} - \frac{1}{4}\hat{\Phi}R_{ws}\right\}.$$

(59)

In the following, we will show that by choosing the integration constants in (48) carefully, the symplectic form and the OPEs of the TsT string theory (40) expressed in the $\hat{X}^\mu$ variable indeed agree with those from the auxiliary AdS$_3$ string theory (59). This suggests that the aforementioned two theories are equivalent even at the quantum level. Consequently, all the rich results of the AdS$_3$ string theory can in principle be mapped to the TsT string theory. For instance, the meaning of (56) and (57) is clear: they are the Noether currents and charges generating the translational symmetry in $\hat{U}$ and $\hat{V}$.

## 4.3 TsT as a canonical transformation

In the previous subsection, we have shown that under the field redefinition (53) the TsT string theory (40) and the auxiliary AdS$_3$ string theory (59) have the same equations of motion and constraints, and hence have the same classical solutions. To fully make use of the map, we still need to establish the equivalence between the two theories at the quantum level. In the following, we will first specify the integration constants of (48) so that the symplectic structure of the TsT string theory (40) in terms of $\hat{X}^\mu$ agree with that from the auxiliary AdS$_3$ string theory (59). Then we will show that the path integral in terms of the phase space variables are equivalent with the said choice of integration constants, and therefore the two apparently different actions (40) and (59) can be obtained by integrating out different choices of momenta.

To do so, let us put the theory on the cylinder and consider the conjugate momenta in both theories

$$p_\mu \equiv 2\pi \frac{\delta S}{\delta(\partial_t x^\mu)}, \qquad p_{\hat{X}^\mu} \equiv 2\pi \frac{\delta \hat{S}}{\delta(\partial_t \hat{X}^\mu)}, \tag{60}$$

where $S$ and $\hat{S}$ are the Lorentzian version of the TsT string action (40) and auxiliary AdS$_3$ worldsheet action (59), respectively. Note that we have absorbed a factor of $2\pi$ in the above definition for convenience. The momenta are given by

$$\begin{aligned}
p_u &= j_0, & p_v &= -\bar{j}_0, \\
p_{\hat{U}} &= \jmath_0 = R_u p_u, & p_{\hat{V}} &= -\bar{\jmath}_0 = R_v p_v, \\
p_{\hat{\Phi}} &= (k-2)\partial_t \phi = p_\phi,
\end{aligned} \tag{61}$$

where we have used the relation (53) and (56). As discussed earlier, using the non-local map (53), the stress tensor in the TsT string theory agrees with that in the auxiliary AdS$_3$ string theory in terms of the phase space variables. In particular, the Hamiltonian can be rewritten in terms of the canonical variables as

$$\begin{aligned}
H &= \frac{1}{2\pi} \int d\sigma \left\{ \frac{p_\phi^2}{2(k-2)} + \frac{k-2}{2}(\partial_\sigma \phi)^2 + p_u \partial_\sigma u - p_v \partial_\sigma v + \frac{2(1+2\lambda \exp(2\phi))}{k \exp(2\phi)} p_u p_v \right\} \\
&= \frac{1}{2\pi} \int d\sigma \left\{ \frac{p_{\hat{\Phi}}^2}{2(k-2)} + \frac{k-2}{2}(\partial_\sigma \hat{\Phi})^2 + p_{\hat{U}} \partial_\sigma \hat{U} - p_{\hat{V}} \partial_\sigma \hat{V} + \frac{2}{k} e^{-2\hat{\Phi}} p_{\hat{U}} p_{\hat{V}} \right\} = \hat{H},
\end{aligned} \tag{62}$$

where $\hat{H}$ denotes the Hamiltonian derived directly from the auxiliary AdS$_3$ worldsheet action (59). Note that the equivalence between the two Hamiltonians does not depend on the choice of integration constants in the field redefinition (53). These integration constants, however, will affect the symplectic form and Poisson brackets if they depend on the states. In terms of the canonical momenta, the symplectic form in the two theories can be written as

$$\omega = \frac{1}{2\pi} \oint d\sigma (\delta x^\mu \wedge \delta p_\mu), \qquad \hat{\Omega} = \frac{1}{2\pi} \oint d\sigma \left( \delta \hat{X}^\mu \wedge \delta p_{\hat{X}^\mu} \right). \tag{63}$$

In order to make the TsT string theory (40) and the auxiliary AdS$_3$ string theory (59) equivalent in a fixed $w$ sector, we need to require that the symplectic forms (63) agree with each other upon the field redefinition (53), i.e.

$$\omega = \hat{\Omega} \, . \tag{64}$$

Matching the symplectic form enables us to use the tools in the auxiliary AdS$_3$ theory to study the TsT string theory. It will be interesting to further understand if there are deeper reasons behind this mapping, which we leave to future study. The above requirement is satisfied if the integration constants are chosen as[5]

$$\eta_0 R_u = \oint \frac{d\sigma}{2\pi w} \hbar[\hat{U} - w\pi, \hat{X}] \, , \qquad \bar{\eta}_0 R_v = -\oint \frac{d\sigma}{2\pi w} \bar{\hbar}[\hat{V} - w\pi, \hat{X}] \, , \tag{65}$$

where we have defined the functionals

$$
\begin{aligned}
\hbar[F, \hat{X}] &\equiv F(\hat{U}) p_{\hat{U}} - \frac{1}{2} F'(\hat{U})((k-2)\partial_\sigma \hat{\Phi} + p_{\hat{\Phi}}) - \frac{k-2}{2k} e^{-2\hat{\Phi}} F''(\hat{U}) p_{\hat{V}} \, , \\
\bar{\hbar}[\bar{F}, \hat{X}] &\equiv \bar{F}(\hat{V}) p_{\hat{V}} - \frac{1}{2} \bar{F}'(\hat{V})(-(k-2)\partial_\sigma \hat{\Phi} + p_{\hat{\Phi}}) - \frac{k-2}{2k} e^{-2\hat{\Phi}} \bar{F}''(\hat{V}) p_{\hat{U}} \, .
\end{aligned}
\tag{66}
$$

The first argument in $\hbar[F, \hat{X}]$ specifies the symmetry parameter, and the second argument specifies the coordinate system. For instance, the expression for $\hbar[f, \tilde{x}]$ is the same as (66) with $F(\hat{U})$ replaced by $f(\tilde{u})$ and $\hat{X} = (\hat{U}, \hat{V}, \hat{\Phi})$ replaced by $\tilde{x} = (\tilde{u}, \tilde{v}, \tilde{\phi})$. Using the relation between the $\hat{X}$ and $\hat{x}$ variables, we have the following relation

$$\hbar[F(\hat{U}), \hat{X}] = \hbar[F(\hat{u}/R_u), \hat{x}] R_u \equiv \left( F p_u - \frac{1}{2} \partial_{\hat{u}} F ((k-2)\partial_\sigma \hat{\phi} + p_\phi) - \frac{k-2}{2k} e^{-2\hat{\phi}} \partial_{\hat{u}}^2 F p_v \right) R_u \, , \tag{67}$$

where in $\hbar[F, \hat{x}]$ the derivative of $F$ is taken with respect to $\hat{x}$. In particular, we can also express the integration constants in terms of the $\hat{x}$ variables as

$$\eta_0 = \oint \frac{d\sigma}{2\pi w} \hbar\left[\frac{\hat{u}}{R_u} - w\pi, \hat{x}\right] \, , \qquad \bar{\eta}_0 = -\oint \frac{d\sigma}{2\pi w} \bar{\hbar}\left[\frac{\hat{v}}{R_v} - w\pi, \hat{x}\right] \, . \tag{68}$$

The zero mode here is reminiscent of the zero mode in Appendix A of [51], where a bulk analysis of the asymptotic symmetry for the double-trace $T\bar{T}$ holography can be found. The zero mode in [51] is a special choice to ensure charge integrability, a condition that can be satisfied by other choices as well. On the other hand, the zero modes in this paper are completely fixed by identifying the worldsheet symplectic structure before and after the deformation.

**Canonical quantization**

We have shown that the field redefinition (53) with the choice of the integration constants (65) preserves the canonical symplectic form, which further implies the equivalence of the Poisson brackets

$$\{x^\mu(\sigma), p_\nu(\sigma')\} = 2\pi \delta_\nu^\mu(\sigma - \sigma') \, , \qquad \{\hat{X}^\mu(\sigma), p_{\hat{X}^\nu}(\sigma')\} = 2\pi \delta_\nu^\mu \delta(\sigma - \sigma') \, . \tag{69}$$

As a consistent check, it is straightforward to verify that the Poisson brackets (69) and the Hamiltonian (62) indeed produce the equation of motion (55) in terms of the $\hat{X}^\mu$ variables. In fact, the equivalence between the string theory (40) after the TsT transformation and auxiliary AdS$_3$ string theory (59) can be preserved at the quantum level. This can be shown in the

---

[5]Here $\hat{U}$ and $\hat{V}$ are not periodic functions of $\sigma$ and the range of the integration is taken to be $[0, 2\pi]$.

canonical quantization. Consider the mode expansion on the constant time slice for the $\hat{X}$ variables,

$$\hat{U}(\sigma) = w\sigma + \sum_{n\in\mathbb{Z}} \hat{U}_n e^{-in\sigma}, \qquad \hat{V}(\sigma) = w\sigma + \sum_{n\in\mathbb{Z}} \hat{V}_n e^{-in\sigma}, \qquad \hat{\Phi}(\sigma) = \sum_{n\in\mathbb{Z}} \hat{\Phi}_n e^{-in\sigma},$$

$$p_{\hat{X}^\mu}(\sigma) = \sum_{n\in\mathbb{Z}} p_{\hat{X}^\mu,n} e^{-in\sigma}, \qquad\qquad \hat{X}^\mu \in \{\hat{U}, \hat{V}, \hat{\Phi}\}, \tag{70}$$

and similarly for the $x^\mu$ variables. To perform canonical quantization, we simply replace the canonical Poisson brackets by commutators with the relation $[,\,] = i\hbar\{,\,\}$. For the $\hat{X}$ variables, the Poisson brackets (69) leads to the commutators

$$[\hat{X}_n^\mu, p_{\hat{X}^\nu,m}] = i\delta_\nu^\mu \delta_{n,-m}, \qquad m,n\in\mathbb{Z}, \tag{71}$$

where we have set $\hbar = 1$ for simplicity. The field redefinition (53) and the integration constants (65) have to be defined in the sense of normal ordering with

$$: p_{\hat{U},n}\hat{U}_{-n} := \begin{cases} p_{\hat{U},n}\hat{U}_{-n}, & n < 0, \\ \hat{U}_{-n}p_{\hat{U},n}, & n \geq 0, \end{cases} \tag{72}$$

and similarly for $p_{\hat{V}}$ and $\hat{V}$. Using these conditions, one can verify that the canonical commutation relations (71) indeed become

$$[x_n^\mu, p_{\nu,m}] = i\delta_\nu^\mu \delta_{n,-m}, \qquad m,n\in\mathbb{Z}, \tag{73}$$

which is the canonical quantization of the Poisson brackets for the TsT strings.

**The OPEs**

We can also proceed with a radial quantization on the plane. In the asymptotic region with $\phi \to \infty$, the OPEs from the action (40) can be written as

$$u(z)j_0(w) \sim \frac{1}{z-w}, \qquad\qquad v(\bar{z})\bar{j}_0(\bar{w}) \sim \frac{1}{\bar{z}-\bar{w}},$$

$$\partial v(z,\bar{z})u(w) \sim -\frac{2\lambda}{k(z-w)}, \qquad\qquad \bar{\partial} u(z,\bar{z})v(w,\bar{w}) \sim -\frac{2\lambda}{k(\bar{z}-\bar{w})}, \tag{74}$$

$$\phi(z,\bar{z})\phi(w,\bar{w}) \sim -\frac{1}{2(k-2)} \log|z-w|^2,$$

where we have ignored terms of order $e^{-2\phi}$ in the last two lines. With the choice of integration constants (65), we have shown that the commutation relation of the TsT string theory (73) is equivalent to that of the auxiliary AdS$_3$ string theory (71). In order to find the OPE in the $\hat{X}^\mu$ variables, it is important to specify the order of operators in the field redefinition. In the following, we keep the order as written in (56) and (53), namely put the rescaling factor $R_u^{-1}$ behind $\hat{u}, j_0$, and similarly for $\hat{V}$ and $\bar{j}_0$. Performing the mode expansion on the Euclidean plane with the commutation relations (71) and normal ordering prescription (72), one can get

$$\hat{\Phi}(z,\bar{z})\hat{\Phi}(w,\bar{w}) = : \hat{\Phi}(z,\bar{z})\hat{\Phi}(w,\bar{w}) : -\frac{1}{2(k-2)} \log|z-w|^2,$$

$$\hat{U}(z)\hat{j}_0(w) = : \hat{U}(z)\hat{j}_0(w) : +\frac{1}{z-w}, \qquad \hat{V}(\bar{z})\bar{\hat{j}}_0(\bar{w}) = : \hat{V}(\bar{z})\bar{\hat{j}}_0(\bar{w}) : +\frac{1}{\bar{z}-\bar{w}}. \tag{75}$$

Therefore the OPEs obtained using the field redefinition (53) indeed agree with that from the auxiliary AdS$_3$ string theory (59),

$$
\begin{aligned}
\hat{\Phi}(z,\bar{z})\hat{\Phi}(w,\bar{w}) &\sim -\frac{1}{2(k-2)}\log|z-w|^2\,, \\
\hat{U}(z)\mathcal{J}_0(w) &\sim \frac{1}{z-w}\,, \qquad \hat{V}(\bar{z})\bar{\mathcal{J}}_0(\bar{w}) \sim \frac{1}{\bar{z}-\bar{w}}\,, \\
\hat{U}(z)\hat{V}(w) &\sim 0\,.
\end{aligned}
\tag{76}
$$

**Path integral and local Lagrangian**

Now we provide a formal derivation of the local Lagrangian in terms of the $\hat{X}$ coordinates, which we have assumed to be the auxiliary AdS$_3$ string action (59). Note that if we directly plug the field redefinition (53) into the action (40), the resulting expression is not (59), but with some extra term. In the path integral, the field redefinition also brings a complicated Jacobian for the measure. This makes it difficult to discuss the relationship of the two theories in the Lagrangian version of the path integral. Instead, let us consider the Hamiltonian version of the path integral in the sector with a fixed winding number $w$

$$
Z_{\text{TsT}} \equiv \int \prod_{\mu} \mathcal{D}x^{\mu}\mathcal{D}p_{x^{\mu}} \exp\left[iS[x,p]\right]\,,
\tag{77}
$$

where $S[x,p]$ is the action (40) written in terms of the phase space variables

$$
S[x,p] = \int dt \oint d\sigma \left(\frac{1}{2\pi}p_{x^{\mu}}\dot{x}^{\mu} - H(x^{\mu},p_{x^{\mu}})\right).
\tag{78}
$$

Firstly, as $x^{\mu},p_{x^{\mu}}$ and $\hat{X},p_{\hat{X}^{\mu}}$ are related by a canonical transformation, the measure of the path integral is kept invariant, namely,[6]

$$
\prod_{\mu}\mathcal{D}x^{\mu}\mathcal{D}p_{x^{\mu}} \equiv \prod_{\mu}\prod_{n\in\mathbb{Z}}dx_n^{\mu}dp_{x^{\mu},-n} = \omega^{\wedge\infty} = \hat{\Omega}^{\wedge\infty} = \prod_{\mu}\mathcal{D}\hat{X}^{\mu}\mathcal{D}p_{\hat{X}^{\mu}}\,.
\tag{79}
$$

This can be viewed as an infinite-dimensional version of the Liouville volume theorem for the canonical transformation driven by $\lambda$. Secondly, we have shown in (62) that if written in terms of the $\hat{X}$ coordinates, the Hamiltonian is just that of AdS$_3$ string theory. Finally, let us focus on the Legendre transformation part of the action (78). Using the field redefinition, we find by direct calculation that the difference is only a total derivative,

$$
\frac{1}{2\pi}\int dt \oint d\sigma \left(p_{x^{\mu}}\dot{x}^{\mu} - p_{\hat{X}^{\mu}}\dot{\hat{X}}^{\mu}\right) = \int dt \frac{d}{dt}\mathcal{B}(t),
\tag{80}
$$

where $\mathcal{B}$ is located at the boundary of the worldsheet and takes the following form

$$
\mathcal{B}(t) = \frac{2\lambda}{k}\left(\eta_0\bar{J}_0 - \bar{\eta}_0 J_0 - \frac{1}{2}\oint \frac{d\sigma}{2\pi}p_u(\sigma)\int_0^{\sigma}d\sigma' p_v(\sigma') + \frac{1}{2}\oint\frac{d\sigma}{2\pi}p_v(\sigma)\int_0^{\sigma}d\sigma' p_u(\sigma')\right).
\tag{81}
$$

Define an operator

$$
U(t) = e^{-i\mathcal{B}(t)}\,,
\tag{82}
$$

---

[6]The volume form of a $2m$ dimensional phase space is given by $\omega^{\wedge m} = \omega \wedge \cdots \wedge \omega$ ($m$ times), where $\omega$ is the symplectic 2-form. Here we have $m \to \infty$.

then the path integral of the TsT string theory can be written as

$$Z_{\text{TsT}} = \int \prod_\mu \mathscr{D}\hat{X}^\mu \mathscr{D}p_{\hat{X}^\mu} \, U_\infty^{-1} \, e^{i\hat{S}[\hat{X}, p_{\hat{X}}]} \, U_{-\infty} \,, \tag{83}$$

where the operator $U$ acts on the past and future boundaries but will not affect the evolution in the middle. After integrating out $p_{\hat{X}^\mu}$ in the path integral, we find that the action in $\hat{X}^\mu$ coordinate is indeed (59) up to terms that act on the past and future boundaries.[7] When the worldsheet manifold is topologically a cylinder, the operators $U_{\pm\infty}$ should be understood as possible dressings of vertex operators inserted at past and future infinity, which will play an important role in the calculation of two-point functions. It is interesting to work out the effect of this dressing more explicitly and furthermore generalize our discussion to generic genus and vertices insertion in general backgrounds. We leave these for future studies.

To summarize, the worldsheet theory (40) on the TsT background can be described by the auxiliary AdS$_3$ string theory (59), at least on flat wordsheet. Using the field redefinition (53), the worldsheet currents, equations of motion, and the stress tensor can all be mapped to each other. With the choice of the integration constants (65), the symplectic form and furthermore the OPEs in the two theories are shown to be equivalent to each other. This suggests a shortcut for studying the TsT transformed string theory: we can map quantities in AdS$_3$ discussed in sec. 3.2 to the TsT transformed theory using the transformation (53). We will use this method to study the asymptotic symmetries in the next section.

# 5 Asymptotic symmetry for strings on TsT deformed AdS$_3$

In this section, we study the asymptotic symmetry for string theory on TsT deformed background (40). On the string worldsheet, asymptotic Killing vectors generate target spacetime diffeomorphisms that preserve the worldsheet equations of motion and stress tensor near the asymptotic boundary. As the nonlocal field redefinition (53) preserves all these asymptotic data, the asymptotic symmetry in the TsT transformed theory can also be obtained from that in the auxiliary AdS$_3$ string theory (59). In section 5.1, we discuss the asymptotic symmetries by applying the idea of [50] directly to the TsT deformed background (40), and show that the asymptotic boundary conditions can be solved by using the non-local map (53). In section 5.2 we discuss the asymptotic symmetry in terms of the $\hat{X}^\mu$ variables, and then in section 5.3 we discuss how the symmetry acts on the original target space coordinates $x^\mu$. We end this section with some comments on the Kac-Moody algebra due to the existence of the internal spacetime.

## 5.1 Asymptotic symmetries from boundary conditions

For the TsT deformed background, the equations of motion are given in (42), and the two conserved currents of the $u, v$ translation are in (43). As in the case of strings on AdS$_3$, we impose the boundary that these currents are finite asymptotically

$$\frac{\bar{\partial}\xi^u}{1 + 2\lambda \exp(2\phi)} - \frac{4\lambda \exp(2\phi)\xi^\phi \bar{\partial}u}{(1 + 2\lambda \exp(2\phi))^2} \sim \mathcal{O}(\exp(-2\phi)),$$

$$\frac{\partial\xi^v}{1 + 2\lambda \exp(2\phi)} - \frac{4\lambda \exp(2\phi)\xi^\phi \partial v}{(1 + 2\lambda \exp(2\phi))^2} \sim \mathcal{O}(\exp(-2\phi)), \tag{84}$$

---

[7]In [66] it was also noticed that the partition function on the plane does not change under the $j^a \wedge j^b$ deformation.

and the variation of equations of motion is of the same order as in (26)

$$\partial\left(\frac{\exp(2\phi)\bar{\partial}\xi^u}{1+2\lambda\exp(2\phi)}+\frac{2\exp(2\phi)\xi^\phi\bar{\partial}u}{(1+2\lambda\exp(2\phi))^2}\right)\sim\mathcal{O}(\exp(-2\phi))\,,$$

$$\bar{\partial}\left(\frac{\exp(2\phi)\partial\xi^v}{1+2\lambda\exp(2\phi)}+\frac{2\exp(2\phi)\xi^\phi\partial v}{(1+2\lambda\exp(2\phi))^2}\right)\sim\mathcal{O}(\exp(-2\phi))\,,\quad(85)$$

$$\partial\bar{\partial}\xi^\phi-\frac{2k\exp(2\phi)(1-2\lambda\exp(2\phi))\xi^\phi\bar{\partial}u\partial v}{(k-2)(1+2\lambda\exp(2\phi))^3}-\frac{k\exp(2\phi)}{k-2}\frac{\bar{\partial}\xi^u\partial v+\partial\xi^v\bar{\partial}u}{(1+2\lambda\exp(2\phi))^2}\sim\mathcal{O}(\exp(-4\phi))\,.$$

The asymptotic symmetries determined by the above boundary conditions can be easily solved by introducing the non-local coordinates as in (47). More explicitly, we have

$$\bar{\partial}\hat{u}=\frac{\bar{\partial}u}{1+2\lambda\exp(2\phi)}\,,\qquad\partial\hat{v}=\frac{\partial v}{1+2\lambda\exp(2\phi)}\,,\qquad\hat{\phi}=\phi\,.\qquad(86)$$

The above relation is preserved by the relation between the variations,

$$\bar{\partial}\xi^u=\bar{\partial}\xi^{\hat{u}}+2\lambda\left(\exp(2\hat{\phi})\bar{\partial}\xi^{\hat{u}}+2\xi^{\hat{\phi}}\exp(2\hat{\phi})\bar{\partial}\hat{u}\right)\,,$$

$$\partial\xi^v=\partial\xi^{\hat{v}}+2\lambda\left(\exp(2\hat{\phi})\partial\xi^{\hat{v}}+2\xi^{\hat{\phi}}\exp(2\hat{\phi})\partial\hat{v}\right)\,,\qquad(87)$$

$$\xi^{\hat{\phi}}=\xi^\phi\,.$$

Using the $\hat{x}$ coordinates, the conditions (84) and (85) are similar to (27) and (26). Thus the asymptotic Killing vectors can be solved as

$$\xi^{\hat{u}}=f(\hat{u})-\frac{k-2}{2k}\exp(-2\phi)\bar{f}''(\hat{v})+\mathcal{O}(\exp(-4\phi))\,,$$

$$\xi^{\hat{v}}=\bar{f}(\hat{v})-\frac{k-2}{2k}\exp(-2\phi)f''(\hat{u})+\mathcal{O}(\exp(-4\phi))\,,\qquad(88)$$

$$\xi^{\hat{\phi}}=-\frac{1}{2}f'(\hat{u})-\frac{1}{2}\bar{f}'(\hat{v})+\mathcal{O}(\exp(-2\phi))\,.$$

There are two subtleties here. First, while the non-local coordinate transformation we have explicitly used in this section is not sensitive to the choice of the zero modes, the resulting asymptotic Killing vectors (88) depend on the non-local coordinates themselves and hence on the zero modes. Second, the windings of $\hat{u}$ and $\hat{v}$ are not integer multiples of $2\pi$, as can be seen from (52). Thus the functions $f(\hat{u})$ and $\bar{f}(\hat{v})$ are not periodic functions of $\hat{u}$ and $\hat{v}$. One way to proceed is to introduce a linear term in $f(\hat{u})$ to take into account the non-trivial boundary condition, an approach similar to the one taken in [51]. On the other hand, as we have already introduced the $\hat{X}$ coordinates which satisfy standard boundary conditions (53), it is more convenient to work in these variables. By varying the map (53), we obtain the relation between the variations

$$\xi^{\hat{u}}=(1+\frac{2\lambda}{wk}\bar{J}_0)\,\xi^{\hat{U}}+\frac{2\lambda}{wk}\hat{U}\delta\bar{J}_0\,,$$

$$\xi^{\hat{v}}=(1+\frac{2\lambda}{wk}\bar{J}_0)\,\xi^{\hat{V}}+\frac{2\lambda}{wk}\hat{V}\delta J_0\,,\qquad(89)$$

$$\xi^{\hat{\phi}}=\xi^{\hat{\Phi}}-\frac{1}{2}\frac{\frac{2\lambda}{wk}\delta J_0}{1+\frac{2\lambda}{wk}J_0}-\frac{1}{2}\frac{\frac{2\lambda}{wk}\delta\bar{J}_0}{1+\frac{2\lambda}{wk}\bar{J}_0}\,.$$

Using these relations, it can be directly shown that the conditions (84) and (85) in terms of $\{\hat{U},\hat{V},\hat{\Phi}\}$ are in the same form of (27) and (26). As a result, the solution to the asymptotic Killing vector is identical to (88) with $\{\hat{u},\hat{v},\hat{\phi}\}$ replaced by $\{\hat{U},\hat{V},\hat{\Phi}\}$. This enables us to proceed with asymptotic Killing vectors in terms of the auxiliary AdS$_3$ variable $\hat{X}$, which we discuss in detail in the following.

## 5.2 The asymptotic symmetry in the $\hat{X}^\mu$ variables

As explained in detail in the previous section, the equations of motion (42) after the TsT transformation is equivalent to (55) in terms of $\hat{X}^\mu$ which is the same as the equations of motion for strings on $AdS_3$ (24). From the field redefinition (53), the asymptotic region with large $\phi$ implies large $\hat{\Phi}$ as well. Then the discussion of the asymptotic symmetry in the $\hat{X}^\mu$ variables are completely parallel to that of AdS$_3$ as summarized in section 3.2, with $\tilde{x}^\mu$ replaced by $\hat{X}^\mu$. By imposing the asymptotic equations of motion similar to (26), the asymptotic Killing vectors can be expressed in terms of two arbitrary functions $F(\hat{U})$ and $\bar{F}(\hat{V})$ as,

$$
\begin{aligned}
\Xi_F &= F(\hat{U})\partial_{\hat{U}} - \frac{k-2}{2k}\exp(-2\hat{\Phi})F''(\hat{U})\partial_{\hat{V}} - \frac{1}{2}F'(\hat{U})\partial_{\hat{\Phi}}\,, \\
\bar{\Xi}_{\bar{F}} &= \bar{F}(\hat{V})\partial_{\hat{V}} - \frac{k-2}{2k}\exp(-2\hat{\Phi})\bar{F}''(\hat{V})\partial_{\hat{U}} - \frac{1}{2}\bar{F}'(\hat{U})\partial_{\hat{\Phi}}\,,
\end{aligned}
\tag{90}
$$

where prime denotes derivative with respect to its argument, and we have omitted the subleading terms. To preserve the periodic boundary conditions (54), the functions $F(\hat{U})$, $\bar{F}(\hat{V})$ should be periodic functions of their respective arguments and thus can be decomposed into Fourier modes

$$
F_m(\hat{U}) = -\exp(im\hat{U})\,, \qquad \bar{F}_m(\hat{V}) = \exp(-im\hat{V})\,.
\tag{91}
$$

As the vectors $\Xi$ only depend on the target spacetime coordinates with state-independent boundary conditions, the commutator between two vectors is simply given by the Lie bracket. Then the generators $\Xi_m \equiv \Xi_{F_m}$ and $\bar{\Xi}_m \equiv \bar{\Xi}_{\bar{F}_m}$ form left and right moving Witt algebra under Lie bracket,

$$
\begin{aligned}
[\Xi_n, \Xi_m] &= i(n-m)\Xi_{n+m}\,, \\
[\bar{\Xi}_n, \bar{\Xi}_m] &= i(n-m)\bar{\Xi}_{n+m}\,, \\
[\Xi_n, \bar{\Xi}_m] &= 0\,.
\end{aligned}
\tag{92}
$$

Now let's calculate the conserved charge corresponding to the symmetry vector $\Xi_F$ and $\xi_F$ in the Hamiltonian formalism. In the following, we focus on the left moving part parameterized by $F(\hat{U})$, whereas discussions on the right moving part are similar. As outlined in section 3.1, at each point on the worldsheet we consider the six-dimensional phase space with coordinates $\{\hat{U}, \hat{V}, \hat{\Phi}, p_{\hat{U}}, p_{\hat{V}}, p_{\hat{\Phi}}\}$. Let $\zeta^I$ denote the tangent vector in the phase space, whose three-dimensional part is given by the asymptotic Killing vector (90), namely,

$$
\zeta^\mu \equiv \{\hat{X}^\mu, \mathcal{J}_F\} = \Xi_F^\mu\,,
\tag{93}
$$

where $\mathcal{J}_F$ generates the transformation (90) on the target spacetime coordinates $\hat{X}^\mu$ via the Poisson bracket. The components of $\zeta$ in the directions of the momenta are determined by the conditions (16) which in this case are given by

$$
\begin{aligned}
\{\mathcal{J}_F, H\} &\sim \mathcal{O}(e^{-2\hat{\Phi}})\,, \\
\{\zeta^I, H\} - \{\{\hat{Q}^I, H\}, \mathcal{J}_F\} &\sim \mathcal{O}(e^{-2\hat{\Phi}})\,.
\end{aligned}
\tag{94}
$$

The meaning of the above two equations is that the symmetry transformation preserves the worldsheet Hamiltonian and equations of motion in the asymptotic region. The specific fall-off condition on the right hand side corresponds to Brown-Henneaux boundary conditions in the auxilliary AdS$_3$ theory as discussed in [50]. The solution to these equations is

$$
\begin{aligned}
\zeta_F^{p_{\hat{U}}} &= -\hbar[F'(\hat{U}), \hat{X}]\,, \\
\zeta_F^{p_{\hat{V}}} &= 0\,, \\
\zeta_F^{p_\Phi} &= -\frac{k}{2}\left(\partial_\sigma \hat{U} + \frac{2}{k}e^{-2\hat{\Phi}}p_{\hat{V}}\right)F''(\hat{U})\,,
\end{aligned}
\tag{95}
$$

where the functional $\hbar$ is defined in (66) which we reproduce here for convenience,

$$
\begin{aligned}
\hbar[F,\hat{X}] &\equiv F(\hat{U})p_{\hat{U}} - \frac{1}{2}F'(\hat{U})((k-2)\partial_\sigma\hat{\Phi} + p_{\hat{\Phi}}) - \frac{k-2}{2k}e^{-2\hat{\Phi}}F''(\hat{U})p_{\hat{V}}, \\
\bar{\hbar}[\bar{F},\hat{X}] &\equiv \bar{F}(\hat{V})p_{\hat{V}} - \frac{1}{2}\bar{F}'(\hat{V})(-(k-2)\partial_\sigma\hat{\Phi} + p_{\hat{\Phi}}) - \frac{k-2}{2k}e^{-2\hat{\Phi}}\bar{F}''(\hat{V})p_{\hat{U}}.
\end{aligned}
\tag{96}
$$

Plugging the variations (93) and (95) into (15), we can obtain the infinitesimal charge

$$
\delta\mathscr{J}_F = \frac{1}{2\pi}\int d\sigma\; \delta\hbar[F(\hat{U}),\hat{X}],
\tag{97}
$$

which is integrable and the resulting finite charge is given by

$$
\mathscr{J}_F = \frac{1}{2\pi}\int d\sigma\; \hbar[F(\hat{U}),\hat{X}].
\tag{98}
$$

Under the mode expansion (91), it is straight forward to verify that the charges $\mathscr{J}_m \equiv \mathscr{J}_{F_m}$ satisfy the Virasoro algebra via the Poisson bracket (69), namely

$$
\begin{aligned}
\{\mathscr{J}_n, \mathscr{J}_m\} &= -i(n-m)\mathscr{J}_{n+m} - in^3\frac{c}{12}\delta_{n,-m}, \\
\{\bar{\mathscr{J}}_n, \bar{\mathscr{J}}_m\} &= -i(n-m)\bar{\mathscr{J}}_{n+m} - in^3\frac{c}{12}\delta_{n,-m}, \\
\{\mathscr{J}_n, \bar{\mathscr{J}}_m\} &= 0,
\end{aligned}
\tag{99}
$$

where the central term is $c = 6(k-2)w \sim 6kw$ in the classical limit. Note that the zero mode charges $\mathscr{J}_0$, $\bar{\mathscr{J}}_0$ generate translations in $\hat{U}$ and $\hat{V}$, respectively.

## 5.3 The asymptotic symmetry for the TsT strings

So far the asymptotic charges (98) have been constructed so that they correspond to the asymptotic Killing vectors (90) in the $\hat{X}$ variables. As shown in the last section, the auxiliary AdS$_3$ string theory is equivalent to the string theory on the linear dilaton background (40) under the field redefinition (53). As the transformations (90) preserve the worldsheet equations of motion and stress tensor asymptotically in the former theory, they preserve those in the later theory as well. Therefore the charges (98) also generate asymptotic symmetries in the TsT string theory (40). Now let us consider the action of these charges on $x^\mu$ which is the physical target spacetime coordinates after the TsT transformation.

Using the Poisson brackets (69) and the field redefinition (53), it is straightforward to work out the Poisson brackets between the charges and the $\hat{x}$ coordinates, which can be written as

$$
\begin{aligned}
\{\hat{u}, \mathscr{J}_F\} &= \mathscr{f}_F(\hat{u}) - \frac{k-2}{2k}\exp(-2\hat{\phi})\bar{\mathscr{f}}_F''(\hat{v}), \\
\{\hat{v}, \mathscr{J}_F\} &= \bar{\mathscr{f}}_F(\hat{v}) - \frac{k-2}{2k}\exp(-2\phi)\mathscr{f}_F''(\hat{u}), \\
\{\phi, \mathscr{J}_F\} &= -\frac{1}{2}\mathscr{f}_F'(\hat{u}) - \frac{1}{2}\bar{\mathscr{f}}_F'(\hat{v}),
\end{aligned}
\tag{100}
$$

where the function $\mathscr{f}_F(\hat{u})$ is given by

$$
\mathscr{f}_F(\hat{u}) = (F(\hat{U}) + \hat{u}\, w_F)R_u, \qquad\qquad\qquad \bar{\mathscr{f}}_F(\hat{v}) = \hat{v}\,\bar{w}_F R_u, \tag{101}
$$

$$
w_F = \{R_u, \mathscr{J}_F\}R_u^{-2} = -\frac{\bar{J}_0\mathscr{J}_{F'}}{1 + \frac{2\lambda}{wk}J_0 + \frac{2\lambda}{wk}\bar{J}_0}\left(\frac{2\lambda}{wkR_u}\right)^2, \qquad\qquad \bar{w}_F = \frac{\mathscr{J}_{F'}}{1 + \frac{2\lambda}{wk}J_0 + \frac{2\lambda}{wk}\bar{J}_0}\frac{2\lambda}{wkR_u}.
$$

The above transformation in the $\hat{x}$ variables is formally a left-moving conformal transformation with symmetry parameter $\ell_F$ accompanied by a rescaling in the right-moving coordinates $\hat{v}$. Note that the transformation (100) indeed takes the general form of (88), with $f(\hat{u}) = \ell_F(\hat{u})$ when we take $\bar{F}(\bar{V}) = 0$. When $\bar{F}(\bar{V}) \neq 0$, it will contribute yet another linear term in $f(\hat{u})$, similar to the appearance of $\hat{v}\,\bar{w}_F R_u$ due to $F(\hat{U})$. To see the action on the TsT coordinates $x^\mu$, it is useful to note that

$$
\begin{aligned}
\{p_u, \mathcal{J}_F\} &= -\hbar[\ell_F'(\hat{u}), \hat{x}],\\
\{p_v, \mathcal{J}_F\} &= -\bar{\hbar}[\bar{\ell}_F'(\hat{v}), \hat{x}],\\
\{p_\phi, \mathcal{J}_F\} &= -\frac{k}{2}\left(\partial_\sigma \hat{u} + \frac{2}{k} e^{-2\phi} p_v\right)\ell_F''(\hat{u}).
\end{aligned}
\tag{102}
$$

Using the coordinate transformation (53) and the above formula, we obtain the following transformation

$$
\begin{aligned}
\{u, \mathcal{J}_F\} &= \ell_F(\hat{u}) - \frac{k-2}{2k}\exp(-2\hat{\phi})\bar{\ell}_F''(\hat{v}) + \frac{2\lambda}{k}\int_0^\sigma d\sigma'\,\bar{\hbar}[\bar{\ell}_F'(\hat{v}), \hat{x}] + \frac{2\lambda}{k}\{\bar{\eta}_0, \mathcal{J}_F\},\\
\{v, \mathcal{J}_F\} &= \bar{\ell}_F(\hat{v}) - \frac{k-2}{2k}\exp(-2\phi)\ell_F''(\hat{u}) - \frac{2\lambda}{k}\int_0^\sigma d\sigma'\,\hbar[\ell_F'(\hat{u}), \hat{x}] + \frac{2\lambda}{k}\{\eta_0, \mathcal{J}_F\},\\
\{\phi, \mathcal{J}_F\} &= -\frac{1}{2}\ell_F'(\hat{u}) - \frac{1}{2}\bar{\ell}_F'(\hat{v}),
\end{aligned}
\tag{103}
$$

where the Poisson brackets appearing in the first two lines are constants given by

$$
\begin{aligned}
\{\eta_0, \mathcal{J}_F\} &= -\oint \frac{d\sigma}{2\pi w}\,\hbar\left[\left(\frac{\hat{u}}{R_u} - w\pi\right)\ell_F'(\hat{u}), \hat{x}\right] + \mathcal{J}_F\frac{1}{wR_u},\\
\{\bar{\eta}_0, \mathcal{J}_F\} &= \oint \frac{d\sigma}{2\pi w}\,\bar{\hbar}\left[\left(\frac{\hat{v}}{R_v} - w\pi\right)\bar{\ell}_F'(\hat{v}), \hat{x}\right].
\end{aligned}
\tag{104}
$$

We note that the symmetry parameter $\ell(\hat{u})$ now contains a term that is linear in the coordinate. One may wonder if the transformation is compatible with the boundary conditions (45). It turns out the shift of the third term in (103) under $\sigma \to \sigma + 2\pi$ cancels the shift from the linear part in $\ell_F$, so that the variation of the coordinates remains periodic. More explicitly, we have

$$
\begin{aligned}
\delta_F u(2\pi) - \delta_F u(0) &= 2\pi w R_u\left(w_F R_u + \frac{2\lambda}{wk}\bar{w}_F \bar{J}_0\right) = 0,\\
\delta_F v(2\pi) - \delta_F v(0) &= 2\pi w R_v R_u \bar{w}_F - \frac{2\lambda}{k}\oint d\sigma\,\hbar[\ell_F'(\hat{u}), \hat{x}] = 0.
\end{aligned}
\tag{105}
$$

One particularly interesting transformation is the zero mode with $F(\hat{U}) = F_0 = 1$, in which case we have $w_F = \bar{w}_F = 0$, both the linear term and the non-local term vanish, and we find that the charge $\mathcal{J}_0$ shifts the coordinates $u$ and $v$ simultaneously,

$$
\{x^\mu, \mathcal{J}_0\}\partial_\mu = -R_u\partial_u + \frac{2\lambda}{wk}J_0\partial_v.
\tag{106}
$$

On the other hand, we expect to find a set of generators that include the translational generators $J_0, \bar{J}_0$, which generate $\partial_u, \partial_v$ respectively. The relation between $\mathcal{J}_0$ and $J_0$ (57) then suggests that we can define the following charges,

$$
J_F \equiv \mathcal{J}_F R_u^{-1} = \oint \frac{d\sigma}{2\pi}\,\hbar[F(\hat{u}R_u^{-1}), \hat{x}], \qquad \bar{J}_{\bar{F}} \equiv \bar{\mathcal{J}}_{\bar{F}} R_v^{-1},
\tag{107}
$$

where we have used the relation (67). Acting on the TsT coordinates, we find

$$\chi_F^\mu \equiv \{x^\mu, J_F\} = \{x^\mu, \mathscr{J}_F\} R_u^{-1} - J_F \frac{2\lambda}{wkR_u} \delta_v^\mu \,, \tag{108}$$

from which we learn that the zero mode charge with $F = 1$ indeed generates translation in $u$. The most general asymptotic charges in the target spacetime are given by

$$J_{F,\bar{F}} = J_F + \bar{J}_{\bar{F}} \,, \tag{109}$$

and they generate the following transformations on the coordinates.

$$\chi^u \equiv \{u, J_{F,\bar{F}}\} = f_{F,\bar{F}}(\hat{u}) - \frac{k-2}{2k} \exp(-2\hat{\phi}) \bar{f}''_{F,\bar{F}}(\hat{v}) + \frac{2\lambda}{k} \int_0^\sigma \bar{\hbar}[\bar{f}'_{F,\bar{F}}(\hat{v}), \hat{x}] + c_{\bar{f}_{F,\bar{F}}} \,,$$

$$\chi^v \equiv \{v, J_{F,\bar{F}}\} = \bar{f}_{F,\bar{F}}(\hat{v}) - \frac{k-2}{2k} \exp(-2\phi) f''_{F,\bar{F}}(\hat{u}) - \frac{2\lambda}{k} \int_0^\sigma \hbar[f'_{F,\bar{F}}(\hat{u}), \hat{x}] + c_{f_{F,\bar{F}}} \,, \tag{110}$$

$$\chi^\phi \equiv \{\phi, J_{F,\bar{F}}\} = -\frac{1}{2} f'_{F,\bar{F}}(\hat{u}) - \frac{1}{2} \bar{f}'_{F,\bar{F}}(\hat{v}) \,,$$

where[8]

$$f_{F,\bar{F}}(\hat{u}) = F(\hat{U}) + (w_F + w_{\bar{F}})\hat{u} \,,$$
$$\bar{f}_{F,\bar{F}}(\hat{v}) = \bar{F}(\hat{V}) + (\bar{w}_F + \bar{w}_{\bar{F}})\hat{v} \,, \tag{111}$$

and

$$c_{\bar{f}_{F,\bar{F}}} = \frac{2\lambda}{wk} \oint \frac{d\sigma}{2\pi} \bar{\hbar} \left[ \left( \frac{\hat{v}}{R_v} - w\pi \right) \bar{f}'_{F,\bar{F}}(\hat{v}), \hat{x} \right] \,,$$

$$c_{f_{F,\bar{F}}} = -\frac{2\lambda}{wk} \oint \frac{d\sigma}{2\pi} \hbar \left[ \left( \frac{\hat{u}}{R_u} - w\pi \right) f'_{F,\bar{F}}(\hat{u}), \hat{x} \right] \,. \tag{112}$$

Acting on the momenta, we have

$$\chi^{p_u} \equiv \{p_u, J_{F,\bar{F}}\} = -\hbar[f'_{F,\bar{F}}(\hat{u}), \hat{x}] \,,$$

$$\chi^{p_v} \equiv \{p_v, J_{F,\bar{F}}\} = -\bar{\hbar}[\bar{f}'_{F,\bar{F}}(\hat{v}), \hat{x}] \,, \tag{113}$$

$$\chi^{p_\phi} \equiv \{p_\phi, J_{F,\bar{F}}\} = -\frac{1}{2} \left( \partial_\sigma \hat{u} + \frac{2}{k} e^{-2\phi} p_v \right) f''_{F,\bar{F}}(\hat{u}) - \frac{1}{2} \left( -\partial_\sigma \hat{v} + \frac{2}{k} e^{-2\phi} p_v \right) \bar{f}''_{F,\bar{F}}(\hat{v}) \,.$$

Note that the asymptotic Killing vector (110) depends on the state and is also non-local on the string worldsheet. It is difficult to see directly how it acts directly on the target spacetime coordinates. Nevertheless, we can show that these vectors are indeed asymptotic Killing vectors in the sense that they preserve the Hamiltonian and the equations of motion. Similar to (94), we find

$$\{J_F, H\} \sim \mathcal{O}(e^{-2\phi}) \,,$$
$$\{\chi^I, H\} - \{\{q^I, H\}, J_F\} \sim \mathcal{O}(e^{-2\phi}) \,. \tag{114}$$

---

[8]The asymptotic Killing vector $\chi^\mu$ with $w = 1$ is similar to (A.7) in [51]. To make the comparison, we can identify $f_{F,\bar{F}}, c_{f_{F,\bar{F}}}$ to $f$ and $c_{\mathscr{L}_f}$ in [51]. In particular, both $f_{F,\bar{F}}$ and $f$ contain a periodic part and a linear term in the coordinates, so that the asymptotic Killing vector still preserves the periodic boundary conditions. The charge $\mathscr{J}_m$ is similar to the 'rescaled' charges, and $J_m$ is similar to the 'unrescaled' charges in [51].

Now let us consider the algebra formed by the charges (107). Under the mode expansion (91), the charges $J_m \equiv J_{F_m}$ form the following algebra via Poisson brackets,

$$\{J_n, J_m\} = -\frac{i(n-m)J_{n+m}}{R_u} - i\frac{c}{12}\frac{n^3\delta_{n,-m}}{R_u^2} - \frac{i(n-m)(\frac{2\lambda}{wk})^2\bar{J}_0 J_m J_n}{R_u(1+\frac{2\lambda}{wk}J_0+\frac{2\lambda}{wk}\bar{J}_0)}\,,$$

$$\{\bar{J}_n, \bar{J}_m\} = -\frac{i(n-m)\bar{J}_{n+m}}{R_v} - i\frac{c}{12}\frac{n^3\delta_{n,-m}}{R_v^2} - \frac{i(n-m)(\frac{2\lambda}{wk})^2 J_0 \bar{J}_m \bar{J}_n}{R_v(1+\frac{2\lambda}{wk}J_0+\frac{2\lambda}{wk}\bar{J}_0)}\,, \qquad (115)$$

$$\{J_n, \bar{J}_m\} = \frac{i(n-m)(\frac{2\lambda}{wk})J_n \bar{J}_m}{1+\frac{2\lambda}{wk}J_0+\frac{2\lambda}{wk}\bar{J}_0}\,.$$

Due to the state-dependence, the modified Lie bracket between two vectors $\chi_F$ and $\chi_G$ parameterized by $F(\hat{U})$ and $G(\hat{U})$ should be defined as

$$[\chi_F, \chi_G]^\mu_{m.L} \equiv \{\chi_G^\mu, J_F\} - \{\chi_F^\mu, J_G\} = \{\{x^\mu, J_G\}, J_F\} - \{\{x^\mu, J_F\}, J_G\}\,, \qquad (116)$$

which can also be written as [67]

$$[\chi_F, \chi_G]_{m.L} = [\chi_F, \chi_G]_{Lie} + \delta_{\chi_F}\chi_G - \delta_{\chi_G}\chi_H\,. \qquad (117)$$

Using the Jacobi identities between $J_F$, $J_G$ and $x^\mu$

$$\{\{x^\mu, J_G\}, J_F\} - \{\{x^\mu, J_F\}, J_G\} = -\{x^\mu, \{J_F, J_G\}\}\,, \qquad (118)$$

we find that the algebra formed by the asymptotic Killing vectors is given by

$$[\chi_F, \chi_G]_{m.L} = -\chi_{\{J_F, J_G\}}\,, \qquad (119)$$

which is isomorphic to the algebra formed by the charges (115).

So far we have worked out the asymptotic symmetries in the target spacetime for the TsT string theory (40) at the classical level. The symmetry can be organized in two ways: the Virasoro generators (98) which generate the transformation (90) in the $\hat{X}$ basis, and the $J_m$ generators which form a nonlinear algebra (115) and generate field dependent diffeomorphism (110) in the $x^\mu$ basis. The zero modes $\mathcal{J}_0$, $\bar{\mathcal{J}}_0$ of the former algebra generate translations of the auxiliary coordinates $\hat{U}$ and $\hat{V}$, whereas the zero modes $J_0$, $\bar{J}_0$ generate translations of the physical coordinates $u$ and $v$. The two sets of charges are related by a field-dependent rescaling (107).

As reviewed in section 2, string theory on the TsT-transformed background (40) is conjectured to be holographically dual to the single-trace $T\bar{T}$ deformed CFT$_2$. For a symmetric orbifold CFT $\mathcal{M}^N/S_N$ with seed CFT $\mathcal{M}$, the single-trace $T\bar{T}$ deformed theory $\mathcal{M}^N_{T\bar{T}}/S_N$ is a symmetric orbifold theory with a (double-trace) $T\bar{T}$ deformed seed theory $\mathcal{M}_{T\bar{T}}$. The Virasoro algebra (99) and the non-linear algebra (115) we found from worldsheet analysis agree with those found from the single-trace $T\bar{T}$ deformed CFT [37], the latter of which was based on the analysis of the double-trace version of $T\bar{T}$ deformation [38] and its holographic dual [51]. In [51], asymptotic symmetry on the TsT-transformed background has also been discussed by studying linearized perturbations in supergravity theory. The appearance of the infinite dimensional symmetry (99) or (115) is compatible with the results of [35], where correlation functions in momentum space is found to take a very simple form. Note that the string background (17) after the TsT transformation is asymptotically flat in the string frame with a linear dilaton, the full theory of which is also conjectured to be holographically dual to little string theory [1]. It will be interesting to understand the implications of the asymptotic symmetries (115) in little string theory and flat holography as well.

## 5.4 The quantum algebra

We have discussed asymptotic symmetries on the string worldsheet at the classical level. We have also shown in the previous section that the symplectic structure and the OPEs in the auxiliary AdS$_3$ string theory (59) are also equivalent to those in the TsT string theory (40). This allows us to proceed with quantization and consider the symmetries at the quantum level as well.

At the quantum level, normal ordering is assumed in the $\mathcal{J}_m$ generators defined in (98). It is more convenient to put the worldsheet theory on the plane. Using the OPEs in the $\hat{X}^\mu$ variables, it is not difficult to verify that the generators $\mathcal{J}_m$ indeed generate the transformation $\Xi_m$ defined in (90) in the large radius region, namely

$$[\hat{X}^\mu, \mathcal{J}_m] = i\Xi_m^{\hat{X}^\mu}, \tag{120}$$

and the commutation relations form a direct sum of two Virasoro algebras

$$
\begin{aligned}
&[\mathcal{J}_n, \mathcal{J}_m] = (n-m)\mathcal{J}_{n+m} + \frac{c}{12}m^3\delta_{n,-m}, \\
&[\bar{\mathcal{J}}_n, \bar{\mathcal{J}}_m] = (n-m)\bar{\mathcal{J}}_{n+m} + \frac{\bar{c}}{12}m^3\delta_{n,-m}, \\
&[\mathcal{J}_n, \bar{\mathcal{J}}_m] = 0.
\end{aligned}
\tag{121}
$$

As discussed around (36), the charges $\mathcal{J}_m$ commute with the worldsheet stress tensor and is thus physical.

Now let us consider the $J_m$ generators defined in (107). There is an ordering ambiguity of the operators at the quantum level. In the following, we always multiply powers of $R_u$ and $R_v$ to the right, namely

$$J_m = \mathcal{J}_m R_u^{-1}, \qquad \bar{J}_m = \bar{\mathcal{J}}_m R_v^{-1}. \tag{122}$$

This prescription is purely due to technical reasons, as it makes it possible to invert the above relation so that we can express $\mathcal{J}_m$ in terms of $J_m$. One can also verify that these charges commute with the worldsheet stress tensor

$$[J_m, T_{ws}] = [J_m, \bar{T}_{ws}] = 0. \tag{123}$$

Using the relation (57), we learn that an eigenstate of $\mathcal{J}_0$ and $\bar{\mathcal{J}}_0$ is also an eigenstate of $J_0$ and $\bar{J}_0$. Denote the eigenvalues of $\mathcal{J}_0$, $\bar{\mathcal{J}}_0$ by $p$, $\bar{p}$, and we have

$$
\begin{aligned}
\mathcal{J}_0|p,\bar{p}\rangle &= p|p,\bar{p}\rangle, & \bar{\mathcal{J}}_0|p,\bar{p}\rangle &= \bar{p}|p,\bar{p}\rangle, \\
J_0|p,\bar{p}\rangle &= \alpha(p,\bar{p})|p,\bar{p}\rangle, & \bar{J}_0|p,\bar{p}\rangle &= \bar{\alpha}(p,\bar{p})|p,\bar{p}\rangle.
\end{aligned}
\tag{124}
$$

The modified eigenvalues can be read from the relation (57) which acting on the states becomes

$$p = \alpha + \frac{2\lambda}{wk}\alpha\bar{\alpha}, \qquad \bar{p} = \bar{\alpha} + \frac{2\lambda}{wk}\alpha\bar{\alpha}. \tag{125}$$

The solution of the above equation is given by

$$
\begin{aligned}
\alpha(x,y) &= \frac{1}{2}(x-y) + \frac{wk}{4\lambda}\left(-1 + \sqrt{1 + \frac{4\lambda}{wk}(x+y) + (\frac{2\lambda}{wk})^2(x-y)^2}\right), \\
\bar{\alpha}(x,y) &= \alpha(x,y) + y - x,
\end{aligned}
\tag{126}
$$

where the functions $\alpha$ and $\bar{\alpha}$ can be viewed as a map from eigenvalues of $\mathcal{J}_0$, $\bar{\mathcal{J}}_0$ to those of $J_0, \bar{J}_0$. The above relation is the same as single-trace $T\bar{T}$ spectrum (4) if we identify

$(p, \bar{p})$ as the undeformed eigenvalues $p = \frac{1}{2}(E(0)R + J(0))$, and $(\alpha, \bar{\alpha})$ as the deformed ones $\alpha = \frac{1}{2}(E(\mu)R + J(\mu))$.

Note that the aforementioned relation between the eigenvalues holds for all eigenstates of the two $U(1)$ generators $\mathcal{J}_0$ and $\bar{\mathcal{J}}_0$. The Virasoro algebra (121) implies that the operators $\mathcal{J}_m$ are ladder operators so that the state $\mathcal{J}_m|p, \bar{p}\rangle$ is an eigenstate of $\mathcal{J}_0, \bar{\mathcal{J}}_0$ with shifted eigenvalues $(p - m, \bar{p})$, and furthermore also an eigenstate of $J_0, \bar{J}_0$ with eigenvalues $(\alpha(p - m, \bar{p}), \bar{\alpha}(p - m, \bar{p}))$. We can promote $\alpha$ to a functional of the operators $\mathcal{J}_0$ and $\bar{\mathcal{J}}_0$, using which we find the following algebra

$$
\begin{aligned}
[J_n, J_m] = J_{n+m} \frac{(n-m)}{1 + \frac{2\lambda}{wk}\bar{J}_0} &+ \frac{\frac{c}{12}m^3 \delta_{n,-m}}{(1 + \frac{2\lambda}{wk}\bar{J}_0)^2} - J_m J_n \frac{2\lambda}{wk} \frac{\bar{\alpha}(\mathcal{J}_0, \bar{\mathcal{J}}_0) - \bar{\alpha}(\mathcal{J}_0 - n, \bar{\mathcal{J}}_0)}{1 + \frac{2\lambda}{wk}\bar{J}_0} \\
&+ J_n J_m \frac{2\lambda}{wk} \frac{\bar{\alpha}(\mathcal{J}_0, \bar{\mathcal{J}}_0) - \bar{\alpha}(\mathcal{J}_0 - m, \bar{\mathcal{J}}_0)}{1 + \frac{2\lambda}{wk}\bar{J}_0}.
\end{aligned}
\tag{127}
$$

To derive the above relation, we have used the definition (122) and the commutators (121). Alternatively, we can also multiply the quantum algebra (127) by $1 + \frac{2\lambda}{wk}\bar{\alpha}(\mathcal{J}_0, \bar{\mathcal{J}}_0)$, so that it becomes

$$
[J_n, J_m] = (n - m)J_{n+m} + \frac{c}{12} \frac{m^3 \delta_{n,-m}}{1 + \frac{2\lambda}{wk}\bar{J}_0} - \frac{2\lambda}{wk}\left(J_n J_m \bar{\alpha}(\mathcal{J}_0 - m, \bar{\mathcal{J}}_0) - J_m J_n \bar{\alpha}(\mathcal{J}_0 - n, \bar{\mathcal{J}}_0)\right). \tag{128}
$$

To understand the relation between the above quantum algebra with the classical one (115), we need to restore $\hbar$ and perform perturbation in $\hbar$. Or alternatively, the classical limit can be obtained by expanding (127) on a state with the expectation value of $\langle \mathcal{J}_0 \rangle \gg m$, $\langle \bar{\mathcal{J}}_0 \rangle \gg m$. Then we have the approximation

$$
\bar{\alpha}(\mathcal{J}_0, \bar{\mathcal{J}}_0) - \bar{\alpha}(\mathcal{J}_0 - m, \bar{\mathcal{J}}_0) \sim m \frac{\partial \bar{\alpha}}{\partial \mathcal{J}_0} = -\frac{m \frac{2\lambda}{wk}\bar{J}_0}{1 + \frac{2\lambda}{wk}J_0 + \frac{2\lambda}{wk}\bar{J}_0}. \tag{129}
$$

Plugging the above relation into (127), and ignoring the ordering in $J_m J_n$, we obtain an expansion of the quantum algebra up to $\mathcal{O}(\hbar)$. The result agrees with (115) if we replace the Poisson bracket by commutator $\{,\} \to -\frac{i}{\hbar}[,]$ with $\hbar = 1$. The aforementioned expansion of our quantum algebra (127) also reduces to the symmetry algebra found in the field-theoretic analysis of double-trace and single-trace $T\bar{T}$ CFT [37,38].

Similar expressions can be obtained for the commutator between the $\bar{J}_m$s. For the mixed commutators, we have

$$
[J_n, \bar{J}_m] = J_n \bar{J}_m \left(1 - \frac{1 + \frac{2\lambda}{wk}\bar{\alpha}(\mathcal{J}_0, \bar{\mathcal{J}}_0 - m)}{1 + \frac{2\lambda}{wk}\bar{J}_0}\right) - \bar{J}_m J_n \left(1 - \frac{1 + \frac{2\lambda}{wk}\alpha(\mathcal{J}_0 - n, \bar{\mathcal{J}}_0)}{1 + \frac{2\lambda}{wk}J_0}\right). \tag{130}
$$

Or equivalently,

$$
J_n \bar{J}_m \left(\frac{1 + \frac{2\lambda}{wk}\bar{\alpha}(\mathcal{J}_0, \mathcal{J}^- - m_0)}{1 + \frac{2\lambda}{wk}\bar{J}_0}\right) - \bar{J}_m J_n \left(\frac{1 + \frac{2\lambda}{wk}\alpha(\mathcal{J}_0 - n, \bar{\mathcal{J}}_0)}{1 + \frac{2\lambda}{wk}J_0}\right) = 0. \tag{131}
$$

## 5.5 The fate of the spacetime Kac-Moody algebra

To end this section, we now turn to the Kac-Moody algebra due to the existence of the internal spacetime in string theory. In the string theory on $AdS_3 \times \mathcal{N}$ background, the worldsheet CFT on the internal manifold $\mathcal{N}$ contains an affine Lie group, generated by currents $K^a$ with the following OPE

$$
K^a(z)K^b(w) = \frac{k'\delta^{ab}/2}{(z-w)^2} + \frac{if_c^{ab}K^c}{z-w} + \cdots, \qquad a, b, c = 1, \cdots, \dim G, \tag{132}
$$

where $G$ is a compact group, $k'$ is the level of the affine Lie algebra $\hat{\mathfrak{g}}_{k'}$, and $f_c^{ab}$ is the structure constant. For instance, when $\mathcal{N} = S^3 \times T^4$, $K^a$ can be taken as either the affine $\widehat{\mathfrak{su}(2)}_{k'}$ currents or the currents on the $T^4$. Our subsequent discussion is universal and does not depend on details of the internal manifold or the choice of the currents. As shown in [47], the worldsheet currents $K^a$ can be used to construct affine Kac-Moody currents in the spacetime CFT. After the TsT transformation, a similar statement can be made to string theory on the auxiliary AdS$_3$ spacetime together with the unaffected internal manifold $\mathcal{N}$. Then we have the Kac-Moody algebra in the spacetime CFT generated by charges $K_n^a$,

$$K_n^a = \frac{1}{2\pi i} \oint dz K^a(z) e^{in\hat{U}(z)}, \tag{133}$$

which satisfies the algebra

$$
\begin{aligned}
[K_n^a, K_m^b] &= i f_c^{ab} K_{n+m}^c + \frac{n\tilde{k}}{2} \delta^{ab} \delta_{n+m,0}, \\
[\mathcal{J}_n, K_m^a] &= -m K_{n+m}^a, \\
[\bar{\mathcal{J}}_n, K_m^a] &= 0,
\end{aligned}
\tag{134}
$$

where $\tilde{k} = k' \oint \frac{dz}{2\pi} \partial \hat{U}$ is the Kac-Moody level in the spacetime CFT. Due to the redefinition (122), the algebra between $K_n^a$ and the charges $J_m$ differ from the last line of the above equation, and becomes

$$
\begin{aligned}
[J_n, K_m^a] &= -K_{n+m}^a \frac{m}{1 + \frac{2\lambda}{wk}\bar{J}_0} + J_n K_m^a \left( 1 - \frac{1 + \frac{2\lambda}{wk}\bar{\alpha}(\mathcal{J}_0 - m, \bar{\mathcal{J}}_0)}{1 + \frac{2\lambda}{wk}\bar{J}_0} \right), \\
[\bar{J}_n, K_m^a] &= \bar{J}_n K_m^a \left( 1 - \frac{1 + \frac{2\lambda}{wk}\alpha(\mathcal{J}_0 - m, \bar{J}_0)}{1 + \frac{2\lambda}{wk}J_0} \right).
\end{aligned}
\tag{135}
$$

The classical limit of the above algebra reduces to the following Poisson bracket

$$
\begin{aligned}
\{J_n, K_m^a\} &= \frac{im}{R_u} \left( K_{m+n}^a + \frac{(\frac{2\lambda}{wk})^2 \bar{J}_0 J_n K_m^a}{1 + \frac{2\lambda}{wk}J_0 + \frac{2\lambda}{wk}\bar{J}_0} \right), \\
\{\bar{J}_n, K_m^a\} &= -\frac{im \frac{2\lambda}{wk} \bar{J}_n K_m^a}{1 + \frac{2\lambda}{wk}J_0 + \frac{2\lambda}{wk}\bar{J}_0}.
\end{aligned}
\tag{136}
$$

It is interesting to note that the Kac-Moody currents also induce translations in the $u, v$ directions which are coordinates on the spacetime CFT. We find the following Poisson brackets

$$
\begin{aligned}
\{u, K_n^a\} &= k_n^a(\hat{u}) + \frac{2\lambda}{k} \int_0^\sigma \hbar[\partial_{\hat{v}} \bar{k}_n^a(\hat{v}), \hat{x}] + \bar{c}_n^a, \\
\{v, K_n^a\} &= \bar{k}_n^a(\hat{v}) - \frac{2\lambda}{k} \int_0^\sigma \left( \hbar[\partial_{\hat{u}} k_n^a(\hat{u}), \hat{x}] + \frac{n K^a e^{\frac{in\hat{u}}{R_u}}}{R_u} \right) + c_n^a, \\
\{\phi, K_n^a\} &= 0,
\end{aligned}
\tag{137}
$$

where

$$
\begin{aligned}
k_n^a(\hat{u}) &\equiv \{\hat{u}, K_n^a\} = -\frac{in(\frac{2\lambda}{wk})^2 \bar{J}_0 K_n^a}{1 + \frac{2\lambda}{wk}J_0 + \frac{2\lambda}{wk}\bar{J}_0} \frac{\hat{u}}{R_u}, \\
\bar{k}_n^a(\hat{v}) &\equiv \{\hat{v}, K_n^a\} = \frac{in\frac{2\lambda}{wk}K_n^a}{1 + \frac{2\lambda}{wk}J_0 + \frac{2\lambda}{wk}\bar{J}_0} \hat{v},
\end{aligned}
\tag{138}
$$

and the constants $c_n^a$, $\bar{c}_n^a$ are given by

$$
\begin{aligned}
c_n^a &= -\frac{2\lambda}{wk}\left(\oint \frac{d\sigma}{2\pi}\hbar\left[\partial_{\hat{u}}k_n^a(\hat{u})\left(\frac{\hat{u}}{R_u}-w\pi\right),\hat{x}\right] + \oint \frac{d\sigma}{2\pi}K^a(\sigma)\left(\frac{\hat{u}}{R_u}-w\pi\right)\frac{ne^{\frac{in\hat{u}}{R_u}}}{R_u}\right), \\
\bar{c}_n^a &= \frac{2\lambda}{wk}\oint \frac{d\sigma}{2\pi}\bar{\hbar}\left[\partial_{\hat{v}}\bar{k}_n^a(\hat{v})\left(\frac{\hat{v}}{R_v}-w\pi\right),\hat{x}\right].
\end{aligned}
\tag{139}
$$

One can check that the transformation (137) still preserves the periodicity of $u$, $v$, despite the fact that it contains linear parts. It is interesting to further understand the implication of this novel transformation on the spacetime coordinates, which we leave for future study.

## Acknowledgments

We would like to thank Luis Apolo, Pengxiang Hao, Ruben Monten, Juntao Wang, and Xianjin Xie for useful discussions.

**Funding information** The work is supported by the national key research and development program of China No. 2020YFA0713000.

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
