# Peer review of "Asymptotic Symmetries in the TsT/T\bar{T} Correspondence"

_SciPost Physics, doi:SciPost Phys. 18, 049 (2025)_

## Round 1 · Referee Report · Anonymous (Referee 1) · 2024-11-7

Report

In this manuscript, the authors investigate the asymptotic symmetries in the context of the TsT/TTbar correspondence. Asymptotic symmetry is a particularly interesting subject in holography. As commented in the manuscript, the coincidence between the asymptotic symmetry of the AdS$_3$ spacetime and the two-dimensional conformal symmetry revealed by Brown and Henneaux is well believed as an indication of AdS$_3$/CFT$_2$ correspondence. The BMS symmetry plays a significant role in the ongoing flat holography program. I think the investigation in this manuscript is quite interesting and provides a non-trivial test of the TsT/TTbar correspondence. I would recommend it for publication, given that the authors address the following concerns:

1) I think the main result from this paper is the discovery of the asymptotic symmetry for the deformed background (40) and its connections to two-dimensional TTbar deformed CFT. Technically, the authors found an interesting non-local field redefinition to realize the TsT transformation. Then they apply this redefinition to the asymptotic symmetries of the undeformed theory. I would suggest the authors to consider that if the asymptotic symmetry of (40) can be derived directly by imposing proper boundary conditions. I think it should be a very meaningful computation which will reveal the robustness of their previous proposal that deriving the asymptotic symmetry from the worldsheet and also provide a consistent check of the proposed non-local field redefinition.

2) I wonder if one can consider a flat limit in the present framework.

3) I wonder if the non-local field redefinition is, in some sense, connected to the cut-off geometry that is dual to TTbar deformation, e.g., 1801.02714.

Recommendation

Ask for major revision

  • validity: high
  • significance: high
  • originality: top
  • clarity: top
  • formatting: excellent
  • grammar: good

Author:  Kangning Liu  on 2024-12-03  [id 5017]

(in reply to Report 1 on 2024-11-07)
Category:
remark
answer to question
correction

We thank the referee for the comments. In the attachment, we provide a detailed response to the questions raised by the referee. For the revisions to our paper related to these questions, please refer to our new submission (v2).

Attachment:

ReplyReferee-1.pdf

---

## Round 1 · Referee Report · Anonymous (Referee 2) · 2024-11-12

Report

This article studies the asymptotic symmetries of type IIB string theory in an asymptotically flat with linear dilaton background that can be obtained by doing a TsT transformation on $AdS_3\times \mathcal{N}$. This backgrounds provides an interesting example of non-AdS holography. The theory in the bulk has a weakly coupled string worldsheet description, which the authors use to derive the asymptotic symmetries. The main points of the paper are:

-one can derive the asymptotic symmetries of a bulk theory, in both Lagrangian and Hamiltonian formalisms, directly from the worldsheet theory, in cases when such theory is trustworthy. This way of computing asymptotic symmetries was introduced by the same authors in a previous article.
-for the specific case of asymptotically flat with linear dilaton background studied in the paper, there exists a field redefinition which maps the worldsheet equations of motion and energy-momentum tensor to those of the worldsheet theory of type IIB string theory in $AdS_3\times\mathcal{N}$
-one can compute from the worldsheet the asymptotic symmetries of $AdS_3$ and use the relation mentioned above to derive those of the linear dilaton background in the initial field basis
-the result is a non-linear algebra which matches the symmetry algebra of a (single-trace) $T\bar{T}$-deformed CFT, which strengthens a previously proposed connection between type IIB string theory in this background and this deformation
-after imposing that the symplectic forms agree between the linear dilaton and auxiliary $AdS_3$ worldsheet theories, the quantum version of the asymptotic symmetry algebra is derived, which is a novel result that was missing in the single-trace $T\bar{T}$ holography literature

I recommend this article to be published in SciPost, after the authors address the following comments and correct the typos in the manuscript:
-the agreement between the symplectic forms in eq 64 is necessary in order to make the TsT deformed theory equivalent to one in auxiliary $AdS_3$ and the authors say that this agreement is something they require ("In order to make the TsT string theory (40) and the auxiliary AdS3 string theory (59) equivalent, we need to require that the symplectic forms (63) agree"), but it would be good if they could clarify why such agreement is motivated. Since the whole theory in the bulk even in the absence of the TsT deformation is not a symmetric orbifold for $k>1$, the perfect matching with single-trace $T\bar{T}$ would be very surprising. However this perfect matching follows from the requirement that the full quantum theory agrees with the auxiliary $AdS_3$ one (not only the quantum algebra, but also the integration constants entering the classical computation are fixed in this way). In case there is some deeper motivation to impose this, the result is indeed something to be further explored, but in case there is no such motivation, the agreement is not surprising, but rooted in the construction and I think this should be stated more clearly
-the full theory dual to type IIB string theory in this background is proposed to be Little string theory compactified to 2d (for large values of the deformation parameter). It would be good if the authors could comment a bit about the implications of these symmetries for LST. Is it expected that these symmetries are indeed symmetries of this theory, namely that LST observables should be constrained by the existence of these infinite dimensional symmetry algebras?
-it is not clear to me how to motivate the choice of fall-offs in eq 88 (are they chosen in order to get the known result in eq 93?)
-there is an extra $f$ index in the first line of eq 9 ($V_f$, but there is no $f$ introduced yet)
-the last line of page 17 "that act on the past and future
boundaries. symmetries in a path integral. They argued the partition" should be rewritten
-there is a bar missing in eq 29 and $g$ should be replaced by $\bar{f}$
-there is a bar missing also in eq 66 and 90
-the expressions for the currents in eq 50 should be corrected (the correct versions are in eq 43)
-the signs in eq 81 should be rechecked

Recommendation

Ask for minor revision

  • validity: high
  • significance: good
  • originality: high
  • clarity: good
  • formatting: good
  • grammar: good

Author:  Kangning Liu  on 2024-12-03  [id 5018]

(in reply to Report 2 on 2024-11-12)
Category:
remark
answer to question
correction

We thank the referee for the comments. In the attachment, we provide a detailed response to the questions raised by the referee. For the revisions to our paper related to these questions, please refer to our new submission (v2).

Attachment:

ReplyReferee-2.pdf

---

## Round 2 · Referee Report · Anonymous (Referee 1) · 2024-12-10

Report

I would like to thank the authors for considering my concerns and
answering the additional questions in the previous report. I am
satisfied with their answers and with the improvements made on the
manuscript, so I would like to recommend the paper for publication.

Recommendation

Publish (meets expectations and criteria for this Journal)

---

## Round 2 · Referee Report · Anonymous (Referee 2) · 2024-12-17

Report

I am satisfied with the answers of the authors and with the modifications of the manuscript.

Recommendation

Publish (meets expectations and criteria for this Journal)

---

## Round 2 · List of Changes

1. We added a new subsection 5.1 and some comments below (101).
  2. We added some comments above section 5.4.
  3. We added some comments below equation (54) and (64).
  4. We added “in the fixed w sector” on page 15 above equation (64).
  5. We added a sentence below equation (94).
  6. Typos in equation (9), (24), (26), (29), (50), (66), (96) are corrected. A sentence in the first lines on page 18 is rewritten.

---

## Editorial Decision

published